# Deep Hierarchical Learning with Nested Subspace Networks for Large Language Models

**Paulius Rauba**
University of Cambridge

**Mihaela van der Schaar**
University of Cambridge

## Abstract

Large neural networks are typically trained for a fixed computational budget, creating a rigid trade-off between performance and efficiency that is ill-suited for deployment in resource-constrained or dynamic environments. Existing approaches to this problem present a difficult choice: training a discrete collection of specialist models is computationally prohibitive, while dynamic methods like slimmable networks often lack the flexibility to be applied to large, pre-trained foundation models. In this work, we propose *Nested Subspace Networks (NSNs)*, a novel architectural paradigm that enables a single model to be dynamically and granularly adjusted across a continuous spectrum of compute budgets at inference time. The core of our approach is to re-parameterize linear layers to satisfy a nested subspace property, such that the function computed at a given rank is a strict subspace of the function at any higher rank. We show that this entire hierarchy of models can be optimized jointly via an uncertainty-aware objective that learns to balance the contributions of different ranks based on their intrinsic difficulty. We demonstrate empirically that NSNs can be surgically applied to pre-trained LLMs and unlock a smooth and predictable compute-performance frontier. For example, a single NSN-adapted model can achieve a 50% reduction in inference FLOPs with only a 5 percentage point loss in accuracy. Our findings establish NSNs as a powerful framework for creating the next generation of adaptive foundation models.

## 1 Introduction

**Motivation**. When we deploy deep learning-based systems in practice, there is a trade-off between two properties: how good the model is (*performance*) and how expensive it is to run (*compute*). Typically, the *larger* the model, the *better* the performance. When using such models at inference (deployment) time, we may want to choose, on-the-fly, how "expensive" vs "fast" a model should be. For instance, we may prefer (i) cheaper models for easier questions in language models; (ii) lower-compute models on phones when battery levels drop; or (iii) more expensive models for safety-critical requests such as medical diagnosis. In this paper, we consider exactly this problem—how to build a single network that can flexibly trade off performance and inference cost at test time.

**Current approaches**. Most popular approaches fall into two main categories. On the one hand, conventional approaches operate by creating smaller, static artifacts from a larger pre-trained model (Cheng et al., 2017), using techniques like network pruning (Han et al., 2015b; Blalock et al., 2020) or knowledge distillation (Gou et al., 2021). More recently, parameter-efficient fine-tuning methods like Low-Rank Adaptation (LoRA) (Hu et al., 2022) have gained popularity for adapting large models, but these also produce a static, low-rank adaptation for a fixed budget. In theory, this approach yields highly optimized models for a specific computational target. In practice, however, this strategy suffers from its static nature; creating a model for a new budget requires repeating the entire, often costly, compression pipeline (Zhu & Gupta, 2017), and it fails to provide the granular, on-the-fly adaptability needed for dynamic environments.

On the other hand, recent methods using dynamic neural networks (Han et al., 2021) operate by designing architectures that can be adjusted at inference time, such as slimmable networks that can drop channels (Yu et al., 2018; Li et al., 2021) or layers (Wu et al., 2018). In theory, these approaches more readily take advantage of a single set of weights to serve multiple budgets. In practice, however, this strategy often comes at the price of much more challenging, specialized training schemes that

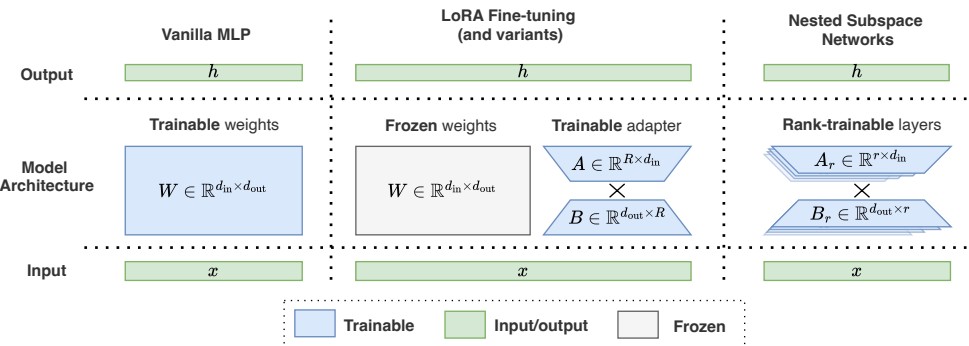

Figure 1: **Illustration of *Nested Subspace Networks*.** NSNs convert linear layers into rank-trainable layers which enable dynamic control over the computational cost (FLOPs) of a forward pass. **Left**: Standard MLP layers that are composed of trainable weights. **Middle**: LoRA fine-tuning which have frozen weights and trainable adapters. **Right**: Nested Subspace Networks replace each linear layer with a *single* pair of shared factor matrices $(A, B)$ defining a rank-trainable layer. The effective weight at rank $r$, $W_r = B_r A_r$, is obtained by using only the first $r$ rows of $A$ and first $r$ columns of $B$. Different operating points (different ranks) therefore correspond to using different prefixes of the same $(A, B)$ This allows for the construction of a compute-performance Pareto frontier at inference.

are typically applied from scratch (Cai et al., 2019). This makes them difficult to apply to the vast ecosystem of existing, pre-trained foundation models, which represent the vast majority of trained and used models today. Further, many of these techniques—with some notable exceptions (Yu & Huang, 2019)—offer only a coarse, discrete set of operating points rather than a smooth, continuous trade-off (Teerapittayanon et al., 2016; Yu et al., 2018).

**Three Desiderata** Can we develop a better approach? Building on the discussion above, we contend that a good solution to the dynamic inference problem should satisfy the following three desiderata.

• *D1: Instant Adaptability*. Instantly trade-off compute and performance at test-time without any additional overhead or expensive fine-tuning procedures in a single neural architecture.

• *D2: Post-Hoc Applicability*. Have the architectural generality to be applied to any pre-trained foundation model and be widely applicable for many classes of models.

• *D3: Granularity*. It should provide a smooth, continuous spectrum of operating points along the compute-performance Pareto frontier, not just a few discrete, pre-determined choices.

In this work, we present an effective method that satisfies these criteria and introduces a new paradigm of flexible model deployment. Our contributions are three-fold.

**Contributions. First**, we introduce Nested Subspace Networks (NSNs), a novel architecture that represents a continuous hierarchy of models within a single set of weights, and we propose a practical uncertainty-aware training objective that makes this hierarchy learnable (Sec. 2). **Second**, we provide theoretical guarantees for granular budget control, showing that our method induces a smooth and predictable performance-compute frontier, even for budgets not explicitly seen during training (Sec. 3). **Third**, we demonstrate the broad utility and effectiveness of NSNs through comprehensive experiments (Sec. 4), including the surgical adaptation of large pre-trained language models, and show that a single adaptive network can match the performance of multiple specialist models.

## 2 NESTED SUBSPACE NETWORKS

**Preliminaries**. Consider a standard feed-forward neural network. A standard linear layer computes the affine transformation $f(\mathbf{x}) = W\mathbf{x} + \mathbf{b}$, where $\mathbf{x} \in \mathbb{R}^{d_{in}}$ is the input vector, the weight matrix is $W \in \mathbb{R}^{d_{out} \times d_{in}}$, and the bias $\mathbf{b} \in \mathbb{R}^{d_{out}}$. The number of parameters in $W$ scales with the product of the

input and output dimensions which becomes a large computational and memory bottleneck. This motivates the need for efficient parametrizations.

**Low-rank factorization** has become a popular approach to mitigating the quadratic cost (Hu et al., 2022), where the full-rank matrix $W$ is approximated with two smaller matrices $W = BA$, where $A \in \mathbb{R}^{R \times d_{\text{in}}}$ and $B \in \mathbb{R}^{d_{\text{out}} \times R}$. Here, $R \ll \min(d_{\text{in}}, d_{\text{out}})$ is a maximum rank. The transformation becomes $f(\mathbf{x}) = (BA)\mathbf{x} + \mathbf{b} = B(A\mathbf{x}) + \mathbf{b}$.

## 2.1 THE NESTED SUBSPACE ARCHITECTURE

NSNs are a class of neural network architectures designed for parameter efficiency and dynamic, post-training adjustment of model capacity. The core principle is to re-parameterize a linear layer with a sequence of low-rank approximations $\{W_r\}_{r=1}^R$ that form a *nested hierarchy*, such that the image of each approximation is a subspace of the next. The architecture is built on the principle of low-rank factorization which we extend to overcome its static limitations and make it applicable to a wide variety of network architectures. Concretely, unlike slimmable networks (Yu et al., 2018), which vary channel width and therefore change intermediate tensor shapes, NSNs only vary the rank of a shared low-rank factorization, so all input–output dimensions of each layer remain fixed and the architecture can be inserted into pre-trained transformers and LLMs without modifying their interfaces or normalization layers.

**Reducing FLOPs**. The low-rank factorization reduces the model's active parameter count and, consequently, its required floating-point operations (FLOPs). This yields a reduction when $r$ is below the break-even point: $2r(d_{\text{in}} + d_{\text{out}}) < 2d_{\text{in}}d_{\text{out}}$, which defines the break-even rank as $\frac{d_{\text{in}}d_{\text{out}}}{d_{\text{in}}+d_{\text{out}}}$.

> **Definition 1 (Nested Subspace Network).** A Nested Subspace Network (NSN) is a neural network architecture that incorporates one or more **NSN layers**. An NSN layer is a linear transformation parameterized by a pair of factor matrices, $A \in \mathbb{R}^{R \times d_{\text{in}}}$ and $B \in \mathbb{R}^{d_{\text{out}} \times R}$, where $R$ is a fixed maximum rank. For a rank $r \in \{1, \ldots, R\}$, the effective weight matrix $W_r \in \mathbb{R}^{d_{\text{out}} \times d_{\text{in}}}$ is constructed from the submatrices $A_r$ (the first $r$ rows of $A$) and $B_r$ (the first $r$ columns of $B$). This is expressed as a sum of the rank-1 outer products, i.e. $W_r := B_r A_r = \sum_{i=1}^r \mathbf{b}_i \mathbf{a}_i$ where $\mathbf{a}_i \in \mathbb{R}^{1 \times d_{\text{in}}}$ is the $i$-th *row* of $A$ and $\mathbf{b}_i \in \mathbb{R}^{d_{\text{out}} \times 1}$ is the $i$-th *column* of $B$.

Note that there is a single pair of factor matrices $(A, B)$ for an NSN layer, and changing the rank $r$ only changes how many of their rows/columns are used to form $W_r$, not the underlying parameters. Appendix D.2 provides an intuitive, worked-out example with a simple matrix. NSNs are a flexible class of models that can operate on model architecture as long as it comprises linear layers. Therefore, it is applicable to models of different sizes, architectures, purposes, with varying inductive biases, etc. A central feature of NSNs is that it naturally gives rise to a fundamental property that we exploit in this work: the *nested subspace property*.

> **Definition 2 (Nested Subspace Property).** The family of weight matrices $\{W_r\}_{r=1}^R$ generated by an NSN layer satisfies the **nested subspace property** if the image of the rank-$r$ transformation is a subspace of the image of the rank-$(r + 1)$ transformation for all $1 \le r < R$:
>
> $$\text{Im}(W_r) \subseteq \text{Im}(W_{r+1}) \quad \forall r \in \{1, \ldots, R-1\}$$
>
> A sufficient condition for this is that $A_{r+1}$ has full row rank for each $r$ (Sec. D.1). This implies the existence of a filtration of vector spaces: $\text{Im}(W_1) \subseteq \text{Im}(W_2) \subseteq \cdots \subseteq \text{Im}(W_R)$.

What does this property mean in practice? NSN layers parametrize *an entire hierarchy of models*. In this hierarchy, due to the nested subspace property, the function class realized by a rank-$r$ model is a strict subset of the function class of a rank-$(r + 1)$ model. Therefore, with the right training scheme, we can choose "which network" we want to employ by deciding on the rank.

However, this approach raises an immediate issue: *how can we train a model that learns a hierarchy of nested sub-models?* A naive approach would be to parametrize the model via low-rank factorization, select a highest rank, train it on the highest rank, and at test time truncate it to the desired rank $r$. While this technically satisfies the nested subspace property, there is no inductive bias to make the models operate along the compute-efficiency Pareto frontier (see Fig 2, implementation details can be found in Appendix B.1.) An alternative approach could be to train *simultaneously* at different ranks

and sum cross entropies within each rank. However, such a naive approach suffers from at least three main issues: (i) it does not take into account the intrinsic difficulty of learning a lower rank model (harder) *versus* a higher rank model (easier); (ii) It results in training instability resulting from large losses in low-rank models; (iii) It is computationally prohibitive to train at all possible ranks. In the next section, we propose an uncertainty-aware training procedure that resolves these challenges.

> **Takeaway.** NSNs create a hierarchy of models within a single network using the *nested subspace property*. The key challenge is training one set of weights to be optimal across this entire hierarchy simultaneously.

## 2.2 TRAINING WITH MULTI-RANK UNCERTAINTY

Our central goal is to learn a single parametrization that simultaneously yields optimal performance for all sub-models. We posit that the failure of naive approaches stems from the differences in intrinsic difficulty of learning different ranks (Sec. 2.1). Therefore, we treat the optimization of each sub-model at different ranks as a multi-task learning problem with varying difficulty levels.

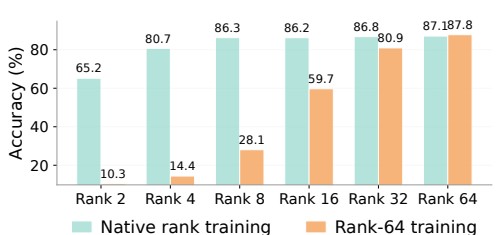

Figure 2: **Comparison of native-rank training and rank truncation for an MLP on CIFAR-10.** The plot compares the accuracy of individually training a model for each specific rank (*Native rank training*) versus training a single model at a high rank (64) and truncating it to lower ranks at test time (*Rank-64 training*). The significant performance gap demonstrates that naively truncating a high-rank model results in poor performance.

**What properties should this optimization objective satisfy?** We seek a weighting mechanism that (i) automatically adapts the relative importance of ranks without per-rank hyperparameter tuning, (ii) is invariant to arbitrary rescalings of the factorization $W = BA$, (iii) guarantees positive weights, and (iv) is cheap enough to apply inside every NSN layer and on every training step. One way to reframe the "difficulty" of a problem is by quantifying the *aleatoric uncertainty* of each task and weighting each task in proportion to that uncertainty (Kendall et al., 2018). We model the aleatoric uncertainty by introducing learnable variance parameters $\sigma_k^2$ for each rank $k$. This variance is assumed to be *heteroskedastic across ranks* (i.e., $\sigma_k^2 \neq \sigma_j^2$ for $k \neq j$) but *homoskedastic within a rank* (i.e., constant for all inputs).

**Modeling assumption.** We consider the classification case (the regression case is analogous). Following Kendall et al. (2018), we use the standard uncertainty-weighted *surrogate* objective in which each rank's cross-entropy is weighted by a learnable scale and regularized by a corresponding log-term. Concretely, for rank $k$ we use the per-rank contribution:

$$\frac{1}{2\sigma_k^2} \mathcal{L}_{\text{CE}}(k) + \log \sigma_k, \tag{1}$$

which serves purely as a weighting-and-regularization surrogate for balancing tasks in classification (i.e., it is not interpreted as a probabilistic likelihood over $\mathcal{L}_{\text{CE}}$).

**Formulating the uncertainty-weighted training objective.** During training, we sample an *anchor rank* $\tilde{R} \leq R$—the maximum rank used at training time—and a *variant rank* $r < \tilde{R}$. Assuming independent uncertainty parameters across ranks, the total objective for a training step is the sum of the two surrogate terms:

$$\left( \frac{1}{2\sigma_{\tilde{R}}^2} \mathcal{L}_{\text{CE}}(\tilde{R}) + \log \sigma_{\tilde{R}} \right) + \left( \frac{1}{2\sigma_r^2} \mathcal{L}_{\text{CE}}(r) + \log \sigma_r \right). \tag{2}$$

We reparameterize the variance by learning its logarithm $s_k = \log(\sigma_k^2)$ and drop the constant factor $\frac{1}{2}$ (Kendall et al., 2018). This results in our final training objective, a function of the shared weights $A$ and $B$ (which define the model weights) and the learnable log-variances:

$$\mathcal{L}_{\text{total}}(A, B, s_{\tilde{R}}, s_r) = \left( \exp(-s_{\tilde{R}}) \mathcal{L}_{\text{CE}}(\tilde{R}) + s_{\tilde{R}} \right) + \left( \exp(-s_r) \mathcal{L}_{\text{CE}}(r) + s_r \right). \tag{3}$$

---

**Algorithm 1** Multi-rank uncertainty-weighted training for NSNs (anchor at maximal rank)

---

**Require:** Dataset $\mathcal{D}$, maximal rank $R$ (anchor), trainable ranks $\mathcal{K} \subseteq \{1, \ldots, R\}$, NSN $f_\theta(X; r)$,
    log-variances $\{s_k = \log \sigma_k^2\}_{k \in \mathcal{K}}$, cross-entropy loss CE, optimizer Opt
1: Initialize $\theta$ and $s_k \leftarrow 0$ for all $k \in \mathcal{K}$
2: **for** each minibatch $(X, Y) \sim \mathcal{D}$ **do**
3:     Sample variant rank $r \sim \mathrm{Uniform}(\mathcal{K} \setminus \{R\})$
4:     Compute anchor logits $Z_R \leftarrow f_\theta(X; R)$, anchor loss $L_R \leftarrow \mathrm{CE}(Z_R, Y)$
5:     Compute variant logits $Z_r \leftarrow f_\theta(X; r)$, variant loss $L_r \leftarrow \mathrm{CE}(Z_r, Y)$
6:     $\mathcal{L}_{\text{anchor}} \leftarrow \exp(-s_R)L_R + s_R, \quad \mathcal{L}_{\text{variant}} \leftarrow \exp(-s_r)L_r + s_r, \quad \mathcal{L}_{\text{total}} \leftarrow \mathcal{L}_{\text{anchor}} + \mathcal{L}_{\text{variant}}$
7:     Opt.zero_grad(); backpropagate $\mathcal{L}_{\text{total}}$ w.r.t. $\theta$ and $\{s_k\}$; Opt.step()
8: **end for**

---

**Why are the exponentials useful in this equation?** It's useful to reason about this from three different perspectives. First, the exponentials are useful because the reparameterization $w_k = e^{-s_k}$ ensures strictly positive weights and produces a strictly convex objective in $s_k$. This is useful, since it provides stable gradient updates. Second, this formulation yields a closed-form optimum $w_k^\star = 1/L_k$. This is useful because we (i) become scale invariant and (ii) have gradient balancing across ranks of different difficulty. Third, this is directly tied to building surrogates that are based on Gaussian regression likelihood for classification settings which are easy to optimize (Kendall et al., 2018). More details in Appendix D.7.

**Insights on the formulated objective.** The uncertainty-weighted surrogate implicitly performs gradient balancing across ranks, since the effective contribution of each term scales with $\exp(-s_k)$. This connects our objective to established approaches for multi-task optimization that seek Pareto-stationary solutions by equilibrating task gradients (Sener & Koltun, 2018), as well as to adaptive weighting schemes such as GradNorm (Chen et al., 2018) and Dynamic Weight Averaging (Liu et al., 2019). This satisfies the required optimization properties (see Appendix D.3 for more discussion) and promotes the well-behaved performance frontier we analyze in Sec. 4.3. The full algorithm is presented in Algorithm 1.

In practice, the inclusion of an anchor rank significantly stabilizes training and helps to learn a better final higher-rank model. Furthermore, we find that introducing a curriculum-learning-based sampling strategy for the variant ranks substantially improves downstream results relative to uniform sampling. Algorithm 4 summarizes the full training procedure. Each iteration evaluates the model at the maximal anchor rank and at a sampled variant rank, combines their losses through rank-specific uncertainty weights, and updates both the shared parameters and the log-variances. This mechanism jointly optimizes all submodels and stabilizes learning across heterogeneous ranks.

> 💡 **Takeaway.** We formalize the joint optimization of different ranks by learning rank-specific homoskedastic variance via a standard uncertainty-weighted surrogate.

## 2.3 Interpreting log variances as a proxy for rank expressiveness

*What has this formulation achieved?* Now, the learned log variances *modulate* the influence of each rank's loss contribution to the objective. High log-variance promotes gradient attenuation by scaling the cross-entropy loss from high-variance ranks. Low log-variance amplifies the penalty loss for cross-entropy losses by increasing its magnitude. In fact, we can directly quantify the gradient contributions based on the loss as:

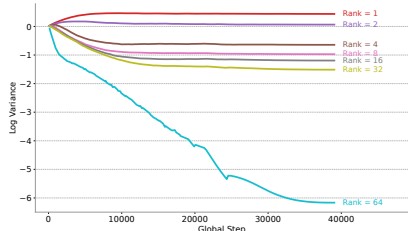

$$\nabla_\theta \mathcal{L}_{\text{total}} = \underbrace{\exp(-s_{\tilde{R}})\nabla_\theta \mathcal{L}_{CE}(\tilde{R})}_{\text{Anchor Contribution}}$$
$$+ \underbrace{\exp(-s_r)\nabla_\theta \mathcal{L}_{CE}(r)}_{\text{Variant Contribution}}$$

Figure 3: **Learned log-variances during training with the multi-rank uncertainty objective** on CIFAR-10 dataset. We train a single model with an anchor and variant ranks and find that higher ranks have lower task-dependent uncertainty during training.

This mechanism allows the learned log-variances to serve as an emergent proxy for the *effective expressiveness* of each rank-specific model. It follows that higher ranks, which possess greater representational capacity, should learn lower corresponding variances. As we demonstrate empirically, this is precisely the behavior we observe in our experiments (Fig. 3).

## 3 PERFORMANCE INTERPOLATION BETWEEN RANKS

A central claim of our work is that NSNs provide granular control over the compute-performance trade-off. This implies not only strong performance at a discrete set of trained ranks but also reliable behavior at *interpolated* ranks that were not explicitly part of the optimization objective. In the absence of theoretical guarantees, one might expect performance to be unpredictable or even collapse between these well-trained points. In this section, we provide the formal underpinnings for the smooth and well-behaved nature of the performance-compute frontier induced by NSNs. We begin by introducing a mild assumption on the structure of the learned weights, from which we derive a formal bound on the interpolation error.

To formalize the notion of a smooth frontier, we must first characterize the structure that our multi-rank uncertainty training (Section 2.2) imposes on the parameterization. We posit that the optimization encourages a natural ordering of basis vectors by importance.

**Assumption 1** (Rank-1 Component Energy Decay). *The training procedure yields a parameterization where the norms of the rank-1 component vectors are monotonically non-increasing with their index. For any indices $i, j$ such that $1 \le i < j \le \tilde{R}$:*

$$\|\mathbf{a}_j\| \le \|\mathbf{a}_i\| \quad and \quad \|\mathbf{b}_j\| \le \|\mathbf{b}_i\|$$

Our training objective motivates this assumption, since it naturally encourages the model to allocate the most salient information to the lowest-indexed basis vectors, as they must be utilized by all nested sub-models. We confirm this assumption empirically (Appendix C.4) and show this is not satisfied in regular training regimes (Appendix C.5).

By extending this result from the model's output to its expected loss, and assuming a standard regularity condition on the loss function, we can establish a bound on the difference in expected performance between any two ranks.

**Proposition 1** (Bound on Interpolation Error). *Let the task loss function $\mathcal{L}(f(\mathbf{x}; r), y)$ be $L_\mathcal{L}$-Lipschitz continuous with respect to its first argument. Let $E(r) = \mathbb{E}_{(\mathbf{x},y)}[\mathcal{L}(f(\mathbf{x}; r), y)]$ be the expected error at rank $r$. For any ranks $r_1 < r_{int} < R$, the difference in expected error is bounded by:*

$$|E(r_{int}) - E(r_1)| \le C \sum_{i=r_1+1}^{r_{int}} \|\mathbf{b}_i\| \|\mathbf{a}_i\|$$

*where $C = L_\mathcal{L} \cdot \mathbb{E}[\|\mathbf{x}\|]$ is a task-dependent constant.*

Proof in Appendix D.4. This result provides a formal guarantee that the variation in model performance is controlled by the cumulative energy of the intermediate basis vectors. This is important because it justifies the use of NSNs for reliable control across a continuous spectrum of computational budgets. We empirically evaluate this claim in Sec. 4.3.

**Takeaway.** With a mild decay assumption on the learned rank-1 components, we can bound the performance change between ranks which ensures that the compute–performance trade-off remains smooth and predictable even for untrained ranks.

## 4 EXPERIMENTAL EVALUATION

Here, we highlight key properties of our method: why do we use two cross-entropy ranks, why our assumptions hold, how we can generalize to interpolated ranks and pre-trained LLMs. [1]

---

[1]The results and code can be found here: https://github.com/pauliusrauba/nested-subspace-networks

## 4.1 IS IT SUFFICIENT TO SIMPLY USE CROSS-ENTROPY ON TWO RANKS?

To better understand the mechanisms behind our proposed training strategy, we conduct a series of ablation studies. Our goal is to isolate the contribution of the core components of our objective function and to validate our design choices against plausible alternatives. We compare several training variations, evaluating their impact not only on the highest-rank model but, more importantly, on the average performance of the resulting lower-rank and interpolated sub-models.

**Setup**. We conduct our evaluation on an image classification task using a standard multi-layer perceptron (MLP) architecture. The dataset is CIFAR-10 throughout but we first map each image to a fixed feature representation using an ImageNet-pretrained backbone and then train only a small classifier on top of these frozen features (more details in Appendix B.3. We assess each method on three metrics: **(i)** the final test accuracy of the highest-rank (anchor) model; **(ii)** the average accuracy across all in-distribution (ID) sub-ranks evaluated during training; and **(iii)** the average accuracy across out-of-distribution (OOD) interpolated ranks not explicitly seen during training. All three variants—Logits Regularization, Residual Orthogonality, and Hidden Regularization—are implemented as independent ablations, each adding exactly one regularizer on top of the same anchor/variant "Two CEs" training setup, and they are never used concurrently. Details are in Appendix B.2. The results are summarized in Table 1.

Table 1: Ablation study of different training objectives. Our core Two CEs formulation (highlighted) dramatically improves the performance of lower-rank (ID) and interpolated (OOD) sub-models. $\tilde{R}$ denotes the anchor rank and $r$ the variant rank. Performance is reported as mean $\pm$ std. dev.

| Method | Key Formulation | Highest Test Acc | Avg. ID Acc | Avg. OOD Acc |
|---|---|---|---|---|
| *Baselines* | | | | |
| CE Only (Anchor) | $\mathcal{L}_{\text{CE}}(\tilde{R})$ | $0.87 \pm 0.00$ | $0.48 \pm 0.00$ | $0.57 \pm 0.00$ |
| One CE + Hard Ortho. | $+ \left\| AA^T - I \right\|_F^2$ | $0.87 \pm 0.00$ | $0.42 \pm 0.00$ | $0.50 \pm 0.00$ |
| *Variations on Joint Training* | | | | |
| Two CEs | $\mathcal{L}_{\text{CE}}(\tilde{R}) + \mathcal{L}_{\text{CE}}(r)$ | $\mathbf{0.88 \pm 0.00}$ | $\mathbf{0.79 \pm 0.00}$ | $\mathbf{0.81 \pm 0.00}$ |
| + Logits Regularization | $+ \left\| \text{logits}(\tilde{R}) - \text{logits}(r) \right\|_2^2$ | $0.87 \pm 0.00$ | $0.64 \pm 0.00$ | $0.64 \pm 0.00$ |
| + Residual Orthogonality | $+ \left\| A_r A_{\text{res}}^T \right\|_F^2$ | $\mathbf{0.88 \pm 0.00}$ | $0.78 \pm 0.00$ | $0.80 \pm 0.00$ |
| + Hidden Regularization | $+ \left\| \mathbf{h}_{\tilde{R}} - \mathbf{h}_r \right\|_2^2$ | $\mathbf{0.88 \pm 0.00}$ | $\mathbf{0.79 \pm 0.00}$ | $0.79 \pm 0.00$ |

**Results**. Our analysis in Table 1 shows that the key to effective sub-model performance is the joint optimization of an anchor and a variant rank ("Two CEs"). This simple objective acts as an implicit regularizer. In contrast, we found that adding explicit regularization terms on top of our proposed objective was either redundant or detrimental. This supports our claim that the joint optimization of multiple ranks is a sufficient mechanism for learning NSNs.

## 4.2 IS THE ENERGY DECAY ASSUMPTION SATISFIED?

**Setup**. We examine the energy-decay property on a Pythia-2.8B model fine-tuned for sequence classification, comparing two variants: a *low-rank* model whose MLP layers are decomposed into rank-1024 factors $A$ and $B$ via SVD, and a standard *dense* model with unmodified linear layers. For each of the 64 MLP weight matrices (32 transformer blocks $\times$ 2 projections), we compute the ordered norms $\|a_i\|_2$, $\|b_i\|_2$ (low-rank) and the row/column norms $\|W_{i,:}\|_2$, $\|W_{:,i}\|_2$ (dense), then report the mean $\pm$ one standard deviation across layers.

**Results**. Results are summarized in Figure 4. We find that the basis vectors in low rank layers consistently show energy, unlike standard dense MLPs.

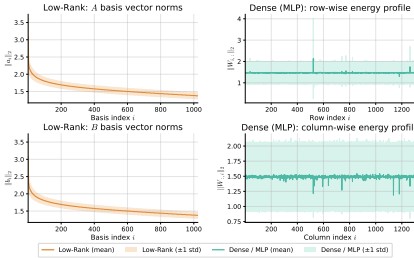

Figure 4: **Energy decay profiles**. The energy decay assumption holds in low-rank linear layers trained with our cross-entropy objective but does not hold in a standard MLP setting.

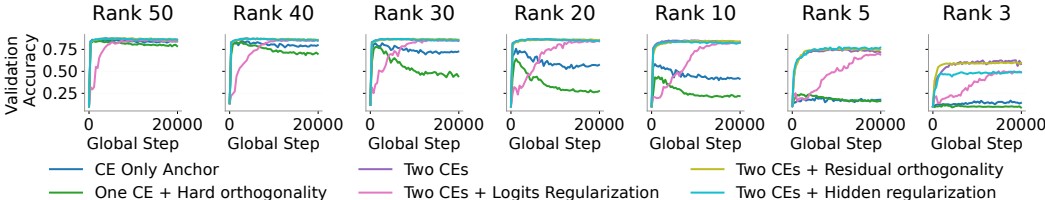

Figure 5: **Training Dynamics at Interpolated Ranks.** Validation accuracy during training for various objective functions, evaluated at ranks not explicitly optimized for. Our proposed method maintains stable learning across all ranks, while simpler baselines exhibit instability and performance collapse.

### 4.3 DOES THIS EMPIRICALLY GENERALIZE TO INTERPOLATED RANKS?

We now empirically validate the theoretical guarantees for smooth interpolation presented in Sec. 3. We ask whether our method yields robust performance even at ranks that were not explicitly part of the optimization process, thereby satisfying our desideratum for granular budget control (Sec 1).

**Setup**. To investigate this, we analyze the performance of models trained with the various objective functions from our ablation study. We train a single model using each objective, which involves a sparse sampling of ranks. We then evaluate the resulting model not only on the anchor rank but also across a wide spectrum of intermediate, interpolated ranks to assess the stability and smoothness of the learned performance curve.

**Results**. Our findings show that the choice of training objective is important for achieving stable interpolation. As illustrated in Figure 5, baseline objectives that do not properly balance the learning dynamics across ranks often result in unstable or collapsing performance at these intermediate points. For example, training with a single cross-entropy objective leads to poor generalization at lower ranks. In contrast, our proposed objective, which jointly trains an anchor and variant rank, produces stable and monotonically improving accuracy curves across the entire hierarchy of ranks throughout training.

### 4.4 DOES THIS EMPIRICALLY GENERALIZE TO PRE-TRAINED LLMS?

We now evaluate the post-hoc applicability of NSNs to large, pre-trained language models (LLMs).

**Adapting Pre-trained Layers.** Our procedure for adapting a pre-trained LLM consists of surgically replacing the standard linear layers within its MLP blocks with NSN layers. A naive approach might be to randomly initialize the NSN factor matrices $A$ and $B$. In practice, however, this method discards the information encoded in the pre-trained weights. To preserve such information in the pre-trained weights, we initialize the NSN factor matrices using Singular Value Decomposition (Appendix D.5).

**Setup**. We apply this adaptation procedure to four publicly available LLMs: *Pythia-2.8B, GPT-Neo-2.7B, Gemma-2B*, and *Qwen2-0.5B*. After replacing and initializing the linear layers in their MLP blocks as described above, we fine-tune each model on a downstream task. For our primary analysis with

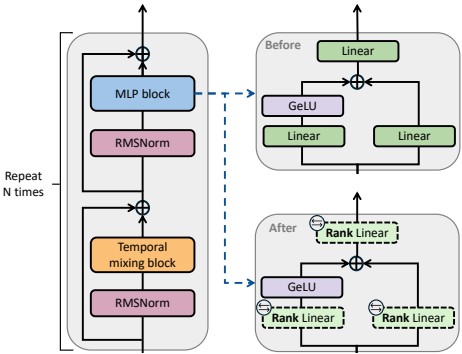

Figure 6: **Example Surgical Changes to linear layers only on Gemma-2B**. The architecture of Gemma-2B. All MLP blocks contain three linear layers with a GeLU activation function. We surgically replace all linear layers with *rank-adaptive* linear layers, initialized $W \approx BA$ via SVD-decomposition

Pythia-2.8B, we use a Natural Language Inference (NLI) benchmark which requires a three-way classification to determine if a premise entails, contradicts, or is unrelated to a given hypothesis. The fine-tuning for all models uses the uncertainty-aware objective described in Section 2.2.

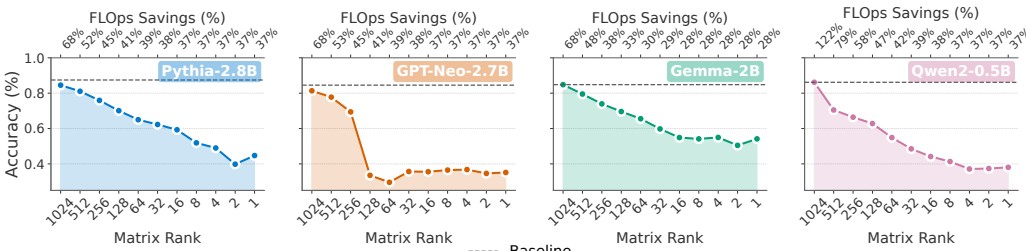

Figure 7: **Accuracy vs rank trade-off for pre-trained LLMs with surgically adapted NSN layers**. We can obtain large reduction in computational cost with minimal decreases in performance.

**Results**. Our results demonstrate that NSNs unlock a smooth and predictable compute-performance frontier for large, pre-trained models. For instance, Pythia-2.8B exhibits a monotonic degradation in accuracy as the rank—and therefore the operational FLOPs—is reduced (Fig. 7). This granular control allows for substantial efficiency gains with a modest performance trade-off; for instance, a 50% reduction in computational cost is achieved with only a 5 percentage point drop in accuracy. We find this behavior is consistent across all four tested language models, where performance drops smoothly as the matrix rank is decreased. This establishes NSNs as an effective method for post-hoc adaptation of foundation models to dynamic inference scenarios.

> **Takeaway.** NSNs can be surgically applied to large, pre-trained foundation models, which allows for smooth and predictable compute-performance trade-offs.

## 5 RELATED WORK

On the one hand, static compression methods aim to create smaller, more efficient, but ultimately fixed models from a larger pre-trained one. Techniques like **network pruning** (Han et al., 2015b; Blalock et al., 2020) and **knowledge distillation** (Gou et al., 2021) excel at this, producing highly optimized artifacts for a specific computational target. However, this approach is fundamentally static; adapting the model to a new computational budget requires repeating the entire, often costly, compression pipeline, failing to provide the on-the-fly adaptability needed for dynamic environments.

On the other hand, **dynamic neural networks** (Han et al., 2021) are designed with inference-time adaptability in mind. **Slimmable networks**, for instance, allow for channels to be dropped dynamically to create sub-networks of varying widths (Yu et al., 2018). While these methods offer the desired adaptability, they typically require specialized and complex training schemes that are applied from scratch (Cai et al., 2019). This makes them difficult to apply to the vast ecosystem of existing, pre-trained foundation models. Furthermore, many such techniques offer only a coarse, discrete set of operating points rather than a smooth, continuous trade-off (Teerapittayanon et al., 2016).

More recently, parameter-efficient fine-tuning (PEFT) methods like **Low-Rank Adaptation (LoRA)** (Hu et al., 2022) have become a popular way to efficiently adapt large models. While LoRA also employs low-rank factorization, its goal is to learn a *single, static update* for a *fixed* rank $r$. It is not designed to be dynamically adjusted at inference time; changing the computational budget would require training a new LoRA adapter with a different rank. In contrast, NSNs leverage a nested low-rank parameterization to enable a single model to be granularly and dynamically adjusted across an entire spectrum of ranks at test time.

Beyond the themes discussed above, Appendix A provides a broader survey situating NSNs within several additional research traditions. It outlines how classical flag-manifold methods study nested subspaces from a geometric perspective, contrasting these representation-space approaches with NSNs' parameter-space formulation. It also reviews dynamic-inference architectures such as Mat-Former, Flextron, and LLAMAFLEX, highlighting how these systems rely on structural slicing or routing rather than the continuous, rank-based hierarchy central to NSNs. Finally, the appendix connects NSNs to adjacent areas—including other low-rank adaptation, adaptive and robust ML,

Table 3: **A comparative analysis against our three desiderata for efficient, adaptable architectures**. Recall the three desiderata where we seek a solution that: **D1:** learns a single, unified *Trade-off Parametrization*[1] that allows for instant *Test-time adaptability*[1]; **D2:** is broadly applicable through *Post-Training Re-parameterization*[2] and exhibits *Architectural Agnosticism*[2] to modify existing pre-trained models; and **D3:** provides granular control by generating a *Smooth trade-off Frontier*[3] across a continuous spectrum of computational budgets. Our proposed method, Nested Subspace Networks (NSNs), is the first to satisfy all five criteria.

| | Method | Example | Trade-off Parametrization[1] | Test-time Adaptability[1] | Post-Training Re-parameterization[2] | Architectural Agnosticism[2] | Smooth Trade-off Frontier[3] |
|---|---|---|---|---|---|---|---|
| LoRA | Standard LoRA | (Hu et al., 2022) | ✗ | ✗ | ✓ | ✓ | ✗ |
| | DyLoRA | (Valipour et al., 2022) | ✗ | ✓ | ✓ | ✓ | ✓ |
| | LoRA-Pruning | (Chen et al., 2023) | ✗ | ✗ | ✓ | ✓ | ✗ |
| Other | Universal Slimmable | (Yu & Huang, 2019) | ✓ | ✓ | ✗ | ✗ | ✓ |
| | Once-for-All (OFA) | (Cai et al., 2019) | ✓ | ✓ | ✗ | ✗ | ✓ |
| | Iterative Magnitude | (Han et al., 2015a) | ✗ | ✗ | ✓ | ✓ | ✗ |
| | Movement Pruning | (Sanh et al., 2020) | ✗ | ✗ | ✗ | ✓ | ✗ |
| | Response-Based KD | (Hinton et al., 2015) | ✗ | ✗ | ✓ | ✓ | ✗ |
| | Self-Distill. (Early Exit) | (Teerapittayanon et al., 2016) | ✗ | ✓ | ✗ | ✓ | ✗ |
| | **Nested Subspace Networks** | **Ours** | ✓ | ✓ | ✓ | ✓ | ✓ |

and test-time adjustment frameworks—clarifying complementarities and highlighting the distinctm echanism how NSNs induce order, controllable capacity and enable adaptation to foundation models.

# 6 DISCUSSION

The dominant paradigm in deploying large models involves creating static artifacts, each trained for a fixed computational budget. This approach is ill-suited for dynamic environments where resource constraints can change on-the-fly. Contrary to the view that this requires training a discrete collection of specialist models—a computationally prohibitive approach—we make a strong case that a single, well-trained dynamic network can effectively and efficiently navigate this trade-off.

We introduced Nested Subspace Networks (NSNs), a novel architectural paradigm that represents a continuous hierarchy of models within a single set of weights. We propose a structural design based on the nested subspace property that has a practical, uncertainty-aware training objective. We show how an entire family of models can be optimized jointly. We further demonstrated that NSNs can be surgically applied post-hoc to large, pre-trained foundation models, unlocking a smooth and predictable compute-performance frontier without requiring training from scratch. This paper presents the first-ever approach to dynamically convert any pre-trained foundation model to a compute-adjustable model with minimal fine-tuning.

*Why and how do NSNs work so well?* We probe this question in our insights experiments (Sec. C). Our analysis reveals that NSNs, regularized by their low-rank structure, converge to different local minima in the loss landscape compared to standard fine-tuning. This means that NSNs find distinct yet highly effective solutions in the loss landscape that allow the family of nested sub-models to work well (Sec. C.3). We empirically verified the foundational nested subspace property, confirming that the vector spaces of lower-rank models are indeed contained within those of higher-rank models, as intended by the architectural design (Sec. C.2). Furthermore, we attempt to understand what happens to the network structure as we change the rank of the network. We find that the low-rank constraint acts as a bottleneck that encourages layers to learn redundant, globally useful functions (Sec. C.1). As capacity increases with higher ranks, layers diverge, adopting more specialized roles. Currently, NSNs shrink or augment all layers to the same rank; we think an interesting–and nontrivial– future work is to develop layer-specific mechanisms for adaptive compute. This requires, however, solving the difficult problem of correlating problem-specific information with layer-specific representational capacity, a problem that has so far attracted little attention. Our findings and insights establish NSNs as a powerful framework for creating the next generation of adaptive foundation models.

**Reproducibility statement** We confirm that all work and figures can be reproduced in the provided github repository.

**Impact Statement** This work advances the field of machine learning and has many downstream applications. While we primarily expect the use of nested subspace networks to be used for advancing positive goals, e.g. making existing systems compute-adaptive and efficient, it is plausible that such techniques could be misused for terminal goals which require the same neural network capabilities. The techniques provided in this paper are designed and tested for language models but can extend well beyond them to any neural network architecture.

**Acknowledgements** We would like to thank the reviewers for their helpful feedback. Claudio Fanconi provided very helpful feedback on an early version of this paper. We thank Atlas Wang for his public engagement with our work which has resulted in us revising our related work. This work was supported by Azure sponsorship credits granted by Microsoft's AI for Good Research Lab.

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

## A EXTENDED RELATED WORK

In this section, we discuss related work which is not immediately related to our discussed problem setup (Sec. 5) yet we find important to cover to better position this work.

**Classical flag-manifold literature** The flag-manifold literature studies nested subspaces to guarantee multi-resolution consistency in representations. Classical foundations include Pennec's formulation of PCA and barycentric subspace analysis on flags, which motivates optimizing over sequences of projectors rather than a single Grassmann subspace (Pennec, 2018); recent geometric toolkits provide algorithms and distances for optimization on flag manifolds (Ye et al., 2022; Nguyen, 2022; Zhu & Shen, 2024; Ye & Lim, 2014). Building on this, Szwagier & Pennec (2025) propose the "flag trick" which replaces a single projector by an average multilevel projector to enforce nestedness across dimensions. Parallel work develops flag-centric representations and statistics (Draper et al., 2014; Mankovich et al., 2022; Ma et al., 2021) and robust principal directions via "flagification" (Mankovich et al., 2024). Mankovich et al. (2025) introduce a flag decomposition that factorizes hierarchical datasets into a hierarchy-preserving flat via a block-modified Gram-Schmidt algorithm. The work presented in this paper is orthogonal to the flag-manifold literature and differs in mechanism, scope, and purpose.

- *In terms of mechanism:* instead of optimizing *data* projections on a flag manifold, NSNs enforce a *parameter-space* filtration inside every linear layer, and couple ranks with an uncertainty-weighted multi-rank objective which yields a smooth compute-performance frontier.
- *In terms of scope*, flag-based approaches treat the nested structure as the object of optimization in representation space and are typically applied to moderate-dimensional linear features, whereas NSNs treat nestedness as an architectural prior that can be injected into arbitrary deep networks (e.g., transformers, CNNs).
- *In terms of purpose*: the *nature* of NSN work is highly practical–we advance the dynamic inference paradigm by introducing an algorithm that obtains different properties from other dynamic inference algorithms. We see that the existing flag-based literature can be applied to enhance our proposed modeling paradigm, but our work does not rely on the explicit setups within the clssical flag-manifold literature.

**Slimmable and Universally Slimmable networks** Slimmable (Yu et al., 2018) and universally-slimmable networks (Yu & Huang, 2019) train a single model that runs at multiple channel widths, enabled by "sandwich"and in-place distillation rules (that this paper shows are unnecessary). In contrast to slimmable networks, whose behavior between trained widths is controlled only empirically through regularizers such as the sandwich rule, the nested subspace structure of NSNs lets us bound the change in expected loss between any two ranks, yielding a theoretically controlled interpolation between compute budgets. Moreover, universally slimmable networks (Yu & Huang, 2019) require width-specific normalization statistics and repeated width sampling during training, which makes them costly and hard to use as post-hoc adapters for large foundation models, whereas NSNs share a single set of parameters and normalization across all ranks and can be added by a short SVD-based fine-tuning phase.

**Further discussion on dynamic inference via nested/elastic architectures** Once-for-all extends this idea by training a supernet whose sub-networks are specialized post-training (Cai et al., 2019). For Transformers, MatFormer nests feed-forward blocks to slice width at inference, relies on supernet training and produces a finite sub-model choice set (Devvrit et al., 2024). Flextron (Cai et al., 2024) converts a pretrained LLM into a nested elastic network and learns routers (static or input-adaptive) that select heads/neurons per budget, after continued training; it provides many sub-models but through discrete head/width choices and routing rather than a single operator with continuous capacity. LLAMAFLEX (Cai et al., 2025) similarly starts from a pretrained LLM and trains a weight-shared, depth/width-slicible architecture and a Gumbel-Softmax router to "train once, deploy many", interpolating between a set of anchor budgets. NSNs, compared to LLAMAFLEX, take a very different approach: we reparameterize linear layers into nested low-rank subspaces and train a single hierarchy of ranks, yielding smooth, theoretically bounded performance–compute curves even at interpolated ranks. Therefore, NSNs are particularly attractive when architecture-agnostic,

training-efficient, and granular control over compute budgets is required, rather than the router-driven depth/width slicing emphasized by LLAMAFLEX. Early-exit and dynamic-depth approaches like BrancyNet(Teerapittayanon et al., 2016), LayerDrop (Fan et al., 2019) or DeeBert (Xin et al., 2020) cut computation by skipping layers or exiting early which changes *where* computation happens, not *how expressive* each linear map is. They also yield discrete exits and require auxiliary heads or training-time regularization. Another close in spirit approach is SortedNet (Valipour et al., 2023) which enforces a generalized "sorted" (partly nested) parameter sharing scheme and trains many discrete sub-models via random sub-model sampling with gradient accumulation. In contrast to all of these, NSNs re-parameterize each linear layer as a single pair of factors whose first $r$ rank-1 components define an exact subspace of the $r+1$ model. We show that jointly optimizing ranks with an uncertainty-weighted two-rank objective gives smooth predictable interpolation across all ranks with theoretical guarantees, that we can employ SVD initialization to allow post-hoc surgical adaptation to pre-trained LLMs (without relying on knowledge-distillation, neural architecture searches or other architecture-specific work), and that this enables clear parameter sharing where the most important information is naturally ordered in the basis vectors based on the order of the ranks (which we show in Appendix C.4).

**Low-rank adaptation literature**    A similar but functionally different literature is the low-rank adaptation and layer-adaptive rank selection literature. LoRA (Hu et al., 2022) and its adaptive variants modify or fine-tune models in low rank, but they target fixed ranks per deployment; AdaLoRA (Zhang et al., 2023) reallocates rank across layers during fine-tuning yet still produces a static configuration at inference. WeLore (Jaiswal et al., 2024) studies why low rank emerges in LLMs (via gradient-Hessian subspace stabilization), then performs one-shot uniform rank projection and an LRC-focused PEFT approach. While it offers strong compression and fine-tuning, it chooses a fixed per-layer rank profile instead of training one model to operate continuously across many ranks at test time. DynaBERT (Hou et al., 2020) provides dynamic width/depth BERT variants through distillation and importance re-writing–again discrete structural sub-networks rather than a single operator with nested rank. Complementary theoretical work analyzes the implicit regularization of overparameterized matrix factorization for matrix completion, showing that gradient flow traverses a hierarchy of invariant manifolds and that the limiting solution transitions from minimum nuclear norm to minimum rank as the connectivity of the observed entries increases (Bai et al., 2024). This is related in spirit to NSNs—both exploit low-rank factorizations trained by gradient methods—but our approach explicitly parameterizes a nested rank hierarchy to shape the compute–performance frontier, rather than relying solely on such connectivity-driven implicit biases. Relative to these lines, NSNs: (i) target rank as the adaptation axis so the function class at rank $r$ is a strict subset of rank $r+1$; (ii) train all ranks jointly with gradient-balancing; (iii) guarantees smooth performance-compute frontiers; (iv) is post-hoc applicable to pre-trained foundation models; and (v) is a standalone model that adapts all weights instead of relying on an adapter on top of frozen weights.

**Adaptive and robust machine learning methods**    Our work can be also seen as being related to the adaptive and robust machine learning literature. Concretely, NSNs provide an adaptavle mechanism to control performance at test-time. Other such models exist within this area. Work on adapting machine learning models at test time is rich, both in terms of looking at re-training them (Lu et al., 2018; Raza et al., 2014; Rabanser et al., 2019; Bayram et al., 2022), detecting errors (Agrahari & Singh, 2022; Gama et al., 2004; Halstead et al., 2022), using adaptive algorithms (Farid et al., 2013; Hulten et al., 2001; Dries & Rückert, 2009) or more robust autonomous approaches for hypothesis-driven adaptation and testing (Rauba et al., 2024a). We find many of these works complementary, whereby the adaptation approaches might benefit from Nested Subspace Network-type architectures. However, more research is needed to create mechanisms how to combine NSN-type models with adaptive ML. For instance, adaptive models could be deployed to select which sub-model to use at test-time, while hypothesis generation algorithms (Xiong et al., 2024) or context-aware testing (Rauba et al., 2024b) can be used to operationally decide what is the minimally performant model required to test a particular set of procedures for cost-sensitive testing. One way to achieve this could be to select the nested subspace model that best satisfies the required criteria after performing model auditing across different sub-models (Rauba et al., 2025) or selecting the cheapest sub-model that still matches the requirements of more expensive models via cascade frameworks (Fanconi & van der Schaar, 2025). Therefore, our work can be easily extended to the adaptive ML literature but does not directly compete against it.

# B EXPERIMENTAL DETAILS

## B.1 DETAILS ON NATIVE-RANK TRAINING VS RANK-TRUNCATION

The experiment presented in Figure 2 aims to contrast two approaches for obtaining models at various computational budgets: (i) native-rank training and (ii) rank truncation. All experiments were conducted on the CIFAR-10 dataset using ImageNet embeddings as input to a Multi-Layer Perceptron (MLP).

**Native Rank**. For the "Native rank training" baseline, we trained a series of independent specialist models. Each model corresponds to a specific rank $r \in \{1, 2, ..., 64\}$. The linear layers of each MLP were parameterized using a low-rank factorization $W = BA$, where the inner dimension was fixed to the target rank $r$. Every model was trained from scratch for 30 epochs using the same hyperparameters to represent the empirical Pareto frontier of performance for a given rank.

**Rank Truncation**. For the "Rank-64 training" comparison, we trained a single model with a maximum rank of $R = 64$. At test time, to evaluate performance at a lower rank $r < R$, we simply truncated the factor matrices $A \in \mathbb{R}^{R \times d_{in}}$ and $B \in \mathbb{R}^{d_{out} \times R}$. Specifically, the truncated weight matrix $W_r$ was constructed using only the first $r$ rows of $A$ and the first $r$ columns of $B$. This ensures the nested subspace property is structurally satisfied, but as Figure 2 shows, this naive approach fails to train the shared parameters to be effective across the hierarchy of ranks.

## B.2 IMPLEMENTATION DETAILS FOR THE ABLATIONS IN TABLE 1

For clarity, we summarize how the three ablation variants in Table 1—*Logits Regularization*, *Residual Orthogonality*, and *Hidden Regularization*—are implemented and how they relate to the core "Two CEs" objective.

**Common setup: anchor / variant ranks and base objective.** In all joint-training variants we select an anchor rank $\tilde{R}$ and a strictly smaller variant rank $r$ from the predefined rank set used throughout the paper. Evaluating the NSN at a given rank $k$ yields logits $f(x; k)$ and the corresponding cross-entropy loss

$$\mathcal{L}_{\mathrm{CE}}(k) = \mathrm{CE}\big(f(x; k), y\big).$$

The base objective (Table 1, row "Two CEs") is

$$\mathcal{L}_{\mathrm{TwoCE}} = \mathcal{L}_{\mathrm{CE}}(\tilde{R}) + \mathcal{L}_{\mathrm{CE}}(r).$$

In the main experiments of Sec. 2.2 we use the uncertainty-weighted surrogate

$$\mathcal{L}_{\mathrm{TwoCE}}^{\mathrm{unc}} = \big(\exp(-s_{\tilde{R}}) \mathcal{L}_{\mathrm{CE}}(\tilde{R}) + s_{\tilde{R}}\big) + \big(\exp(-s_r) \mathcal{L}_{\mathrm{CE}}(r) + s_r\big),$$

with learned log-variances $s_{\tilde{R}}$ and $s_r$. All ablations keep this structure fixed and differ only by adding a single extra regularizer. Each row in Table 1 corresponds to a separate training run; we never activate multiple additional regularizers at once.

**Logits Regularization ("+ Logits Regularization").** This variant encourages the logits at ranks $\tilde{R}$ and $r$ to match:

$$\mathcal{L}_{\mathrm{logit}} = \big\| f(x; \tilde{R}) - f(x; r) \big\|_2^2,$$

implemented as MSE with the anchor logits treated as a fixed target (no gradient through $f(x; \tilde{R})$). Introducing a log-variance $s_{\mathrm{logit}}$, the objective becomes

$$\mathcal{L}_{\mathrm{TwoCE+Logits}} = \mathcal{L}_{\mathrm{TwoCE}}^{\mathrm{unc}} + \tfrac{1}{2} \exp(-s_{\mathrm{logit}}) \mathcal{L}_{\mathrm{logit}} + \tfrac{1}{2} s_{\mathrm{logit}}.$$

This matches the `ce_with_consistency` branch in the code.

**Residual Orthogonality ("+ Residual Orthogonality").** Each NSN layer has basis matrix $A \in \mathbb{R}^{R \times d_{\mathrm{in}}}$, where row $i$ represents the $i$-th rank-1 direction. Given $\tilde{R} > r$, we decompose

$$A_r = A_{1:r,:}, \qquad A_{\mathrm{res}} = A_{r+1:\tilde{R},:}.$$

We penalize overlap between these subspaces via

$$\mathcal{L}_{\text{ortho}} = \sum_{\text{NSN layers}} \left\| A_r A_{\text{res}}^\top \right\|_F^2.$$

With log-variance $s_{\text{ortho}}$, the objective is

$$\mathcal{L}_{\text{TwoCE+ResOrtho}} = \mathcal{L}_{\text{TwoCE}}^{\text{unc}} + \exp(-s_{\text{ortho}}) \mathcal{L}_{\text{ortho}} + s_{\text{ortho}}.$$

This corresponds to the `ce_orthogonality` implementation.

The baseline "One CE + Hard Ortho." in Table 1 instead uses only $\mathcal{L}_{\text{CE}}(\tilde{R})$ together with a global orthogonality penalty $\|AA^\top - I\|_F^2$ enforcing approximate row-orthonormality (implemented in `ce_orthogonality_hard`).

**Hidden Regularization ("+ Hidden Regularization").** For each NSN layer we obtain pre-activation hidden representations at ranks $\tilde{R}$ and $r$:

$$h_{\tilde{R}}^{(\ell)} \in \mathbb{R}^{B \times d_\ell}, \qquad h_r^{(\ell)} \in \mathbb{R}^{B \times d_\ell}.$$

We normalize each along the feature dimension,

$$\hat{h}_{\tilde{R}}^{(\ell)} = \text{normalize}(h_{\tilde{R}}^{(\ell)}), \qquad \hat{h}_r^{(\ell)} = \text{normalize}(h_r^{(\ell)}),$$

and define a consistency loss

$$\mathcal{L}_{\text{feat}} = \sum_\ell \left\| \hat{h}_{\tilde{R}}^{(\ell)} - \hat{h}_r^{(\ell)} \right\|_2^2.$$

With log-variance $s_{\text{feat}}$, the full objective is

$$\mathcal{L}_{\text{TwoCE+Hidden}} = \mathcal{L}_{\text{TwoCE}}^{\text{unc}} + \tfrac{1}{2}\exp(-s_{\text{feat}}) \mathcal{L}_{\text{feat}} + \tfrac{1}{2} s_{\text{feat}}.$$

This matches the `ce_with_feature_consistency` branch in the code.

**Summary.** All rows in the "Variations on Joint Training" block share the same NSN architecture and the same anchor/variant rank selection. The base "Two CEs" loss uses cross-entropies at the two ranks; each ablation adds exactly one additional regularizer with its own uncertainty weight. Each variant is trained independently and evaluated separately.

### B.3 DETAILS ON CIFAR-10 EMBEDDINGS AND THE MLP SETUP

The *dataset* is CIFAR-10 throughout but we first map each image to a fixed feature representation using an ImageNet-pretrained backbone and then train only a small classifier on top of these frozen features.

**Backbone feature extractor.** We use the `torchvision` implementation of ResNet-18 with ImageNet-1K pre-trained weights (`ResNet18_Weights.IMAGENET1K_V1`) as a fixed feature extractor. We remove the final classification layer and keep the network up to the global average pooling stage, yielding a mapping $\phi : \mathbb{R}^{3 \times 32 \times 32} \to \mathbb{R}^{512}$ where $\phi(x)$ is a 512-dimensional feature vector. CIFAR-10 images are resized to $224 \times 224$ and normalized with standard ImageNet mean and variance before being passed through $\phi$. For each split (train/test), we precompute and store pairs $(\phi(x_i), y_i)$. The ResNet backbone is kept in evaluation mode and never updated during any of the NSN experiments; gradients are disabled for all its parameters.

For the *Base MLP* baselines, both linear layers are standard dense linear transformations. For the *NSN* ("Low Rank Layer") models, *both* linear layers are implemented as NSN layers (Definition 1), i.e., they use the nested low-rank factorization with a shared maximum rank $R$ and can be evaluated at any active rank $r \leq R$. No other part of the network (in particular, the ResNet-18 backbone) is re-parameterized or modified by NSNs. Thus, the CIFAR-10 experiments should be understood as: (i) CIFAR-10 images $\to$ frozen ImageNet-pretrained ResNet-18 $\to$ 512-D features, and (ii) all subsequent trainable layers in the MLP classifier are NSN (or dense) layers as specified in Table 1.

## C  ADDITIONAL EXPERIMENTS

### C.1  INTER-LAYER WEIGHT SIMILARITY VS. MATRIX RANK

**Objective.**   This experiment investigates the relationship between the representational capacity of layers and their functional roles within the network. The core question is whether increasing layer capacity (i.e., matrix rank) encourages layers to learn more specialized, distinct functions or more similar, redundant ones. The guiding hypothesis is that a low-rank constraint acts as an informational bottleneck, forcing layers to learn redundant, globally useful functions, while increasing the rank enables and encourages functional specialization.

**Methodology.**   For a trained Nested Subspace Network, we analyzed the weight matrices of the MLP layers at various ranks. For each rank $r$ from 1 to 1024, we reconstructed the effective weight matrix $W_r$ for each layer. We then computed the pairwise cosine similarity between the weight matrices of all layers in the network. The average of these pairwise similarities was then plotted against the matrix rank to observe the overall trend.

**Results and Interpretation.**   The results, shown in Figure 8, confirm the hypothesis. At very low ranks, the average inter-layer similarity is high, indicating that the network's layers learn functionally similar and redundant representations. As the matrix rank and thus the layer capacity increase, the average cosine similarity between layers steadily decreases, approaching zero at the highest ranks. This suggests that with greater representational freedom, layers diverge to assume more specialized roles within the network. The low-rank constraint effectively regularizes the network, forcing layers to cooperate on learning general features, while higher ranks allow for a more distributed and specialized division of labor.

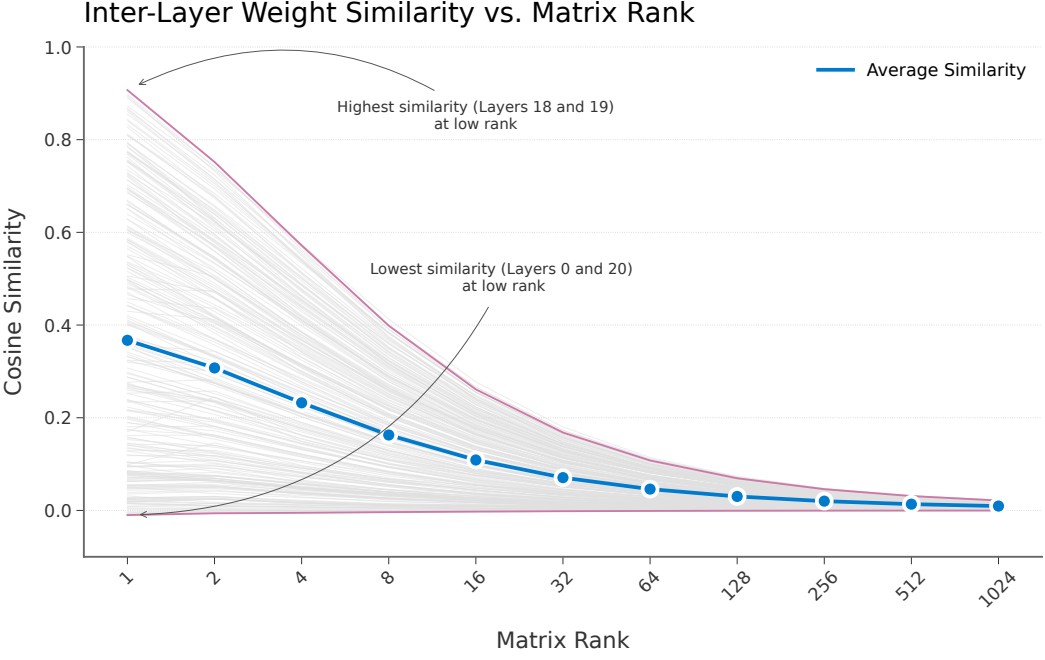

Figure 8: Inter-layer weight similarity as a function of matrix rank. As layer capacity (rank) increases, the average cosine similarity between layers decreases. This suggests that layers transition from learning redundant, globally useful functions at low ranks to more specialized roles at high ranks.

### C.2  EMPIRICAL VERIFICATION OF THE NESTED SUBSPACE PROPERTY

**Objective.**   The central design of Nested Subspace Networks relies on the nested subspace property, where the function computed at a given rank is a strict subspace of the function at any higher rank. This experiment was designed to empirically verify if this theoretical property holds in practice after

training. The key question is: does the vector space spanned by a lower-rank weight matrix truly lie inside the vector space of a higher-rank matrix from the same trained layer?

**Methodology.** To quantify the degree of subspace containment, we used a three-step procedure for each trained NSN layer:

1. **Reconstruct Weights:** For a pair of ranks, $r_{\text{small}}$ and $r_{\text{large}}$, we reconstructed their effective weight matrices, $W_{r_{\text{small}}}$ and $W_{r_{\text{large}}}$.

2. **Find Orthonormal Bases:** We performed Singular Value Decomposition (SVD) on each weight matrix ($W_r = U_r \Sigma_r V_r^T$) to find an orthonormal basis for its column space. The first $r$ columns of the resulting $U_r$ matrix form this basis.

3. **Calculate Containment Score:** We computed a containment score to measure the extent to which the smaller subspace is contained in the larger one. The score is defined as the normalized Frobenius norm of the projection of the smaller basis onto the larger basis:

$$\text{score}(r_{\text{small}}, r_{\text{large}}) = \frac{1}{r_{\text{small}}} \|U_{r_{\text{large}}}^T U_{r_{\text{small}}}\|_F^2$$

A score of 1.0 indicates that the smaller subspace is perfectly contained within the larger one.

**Results and Interpretation.** The results are visualized in the heatmap in Figure 9. The upper triangle of the matrix, where $r_{\text{large}} \geq r_{\text{small}}$, shows scores that are consistently 1.0 or very close to it. This empirically confirms that the vector space of a lower-rank model is indeed a nested subspace of any higher-rank model after training. The lower triangle, where $r_{\text{large}} < r_{\text{small}}$, shows scores significantly less than 1.0. This asymmetry is expected and acts as a sanity check, confirming that a higher-dimensional space cannot be fully contained within a lower-dimensional one.

## C.3 CONVERGENCE ANALYSIS OF LOW-RANK VS. STANDARD FINE-TUNING

**Objective.** This experiment investigates whether a model trained with Nested Subspace layers converges to the same solution in the weight space as a model trained with standard fine-tuning. The hypothesis is that the two models will find different solutions, as the low-rank structure of NSNs acts as a form of regularization that guides the optimization process toward a different local minimum.

**Methodology.** To compare the final learned weights, a standard model was fine-tuned on the task, and a separate NSN-equipped model was trained using our proposed multi-rank objective. For the NSN model, the effective weight matrix for each layer was reconstructed at various ranks. We then computed the cosine similarity between the weight matrix of a layer from the standard fine-tuned model and the corresponding reconstructed matrix from the NSN model. This comparison was performed for all MLP layers, which were grouped into early (0-10), middle (11-20), and late (21-31) stages of the network to observe depth-dependent trends.

**Results and Interpretation.** As shown in Figure 10, the weight matrices of the NSN model do not converge to the same solution as the standard fine-tuned model. The cosine similarity increases with the rank, but even at the highest rank (1024), the similarity is only around 85

This result supports the hypothesis that the nested low-rank structure imposes a regularization effect. By constraining the possible solutions to lie within pre-defined low-rank subspaces, the training process is guided to a different local minimum in the loss landscape than standard, unconstrained fine-tuning. This suggests that NSNs discover a different, yet highly effective, set of parameters for solving the task.

## C.4 VERIFYING ENERGY DECAY ASSUMPTION

To empirically evaluate Assumption 1, we perform an empirical investigation on a chosen language model and multiple layers within this model. Specifically, we directly inspect the learned basis vectors of every `DynamicLowRankLinear` layer in the NSN-adapted GPT-NeoX model (chosen for convenience).

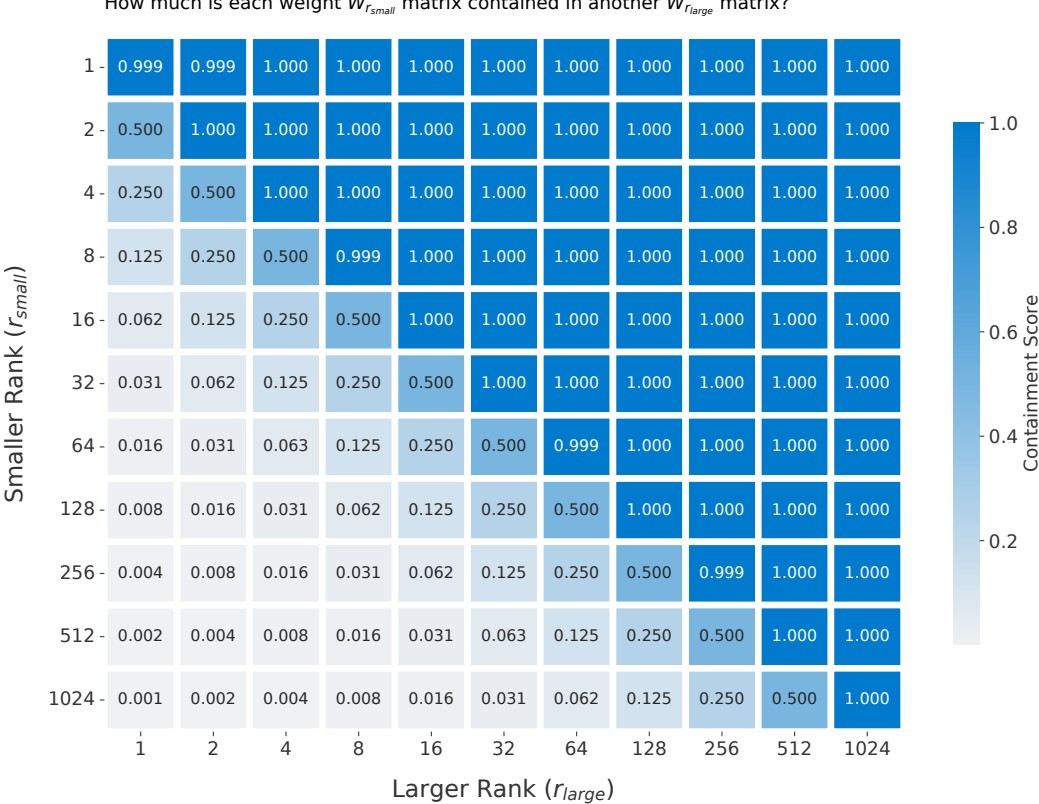

Figure 9: Heatmap of subspace containment scores between weight matrices of different ranks. The score measures the extent to which the column space of a lower-rank matrix ($r_{\text{small}}$) is contained within that of a higher-rank matrix ($r_{\text{large}}$). A score of 1.0 (dark blue) indicates full containment. The results empirically validate the foundational nested subspace property of the trained network.

**Setup**. Each such layer is parameterized as $W_r = \sum_{i=1}^r \mathbf{b}_i \mathbf{a}_i$, where the rows of $A$ provide the $\mathbf{a}_i$ components and the columns of $B$ provide the $\mathbf{b}_i$ components. For every layer, we compute the Euclidean norms $\|\mathbf{a}_i\|_2$ and $\|\mathbf{b}_i\|_2$ across all rank-1 components $i = 1, \ldots, R$ and test whether these sequences are monotonically non-increasing. This monotonicity captures the "energy decay" structure posited by Assumption 1, which states that earlier basis components should contain more salient functional information than later ones. For each layer, we report: the maximum rank $R$, whether monotonicity holds (T/F), the number of violations, and the magnitude of the first and last component norms. This provides a layer-by-layer diagnostic of how strongly the trained model conforms to the nested subspace ordering implied by our theoretical analysis.

**Takeaway**. The results reveal that most layers exhibit a clear decaying trend in the norms of their rank-1 components, even when strict monotonicity is not perfectly satisfied. Violations are typically small and localized, while the overall decrease between the first and last components remains substantial. In total, they constitute 0.04% of all basis vector orderings which is extremely negligible; and we interpret this as noise in the optimization process. This provides strong empirical support for Assumption 1: the optimization process tends to allocate high-energy, high-importance directions to early basis indices, enabling the smooth interpolation behavior and predictable compute–performance trade-offs that NSNs rely on.

To make the picture more precise, we also added a layer-by-layer table reporting how often the assumption is locally violated. These violations happen occasionally—typically one or two indices within a layer. We estimate this amounts to only about one percent of all basis vectors. Because they are sparse and small, and given how consistently (about 99% of all basis vectors) this assumption

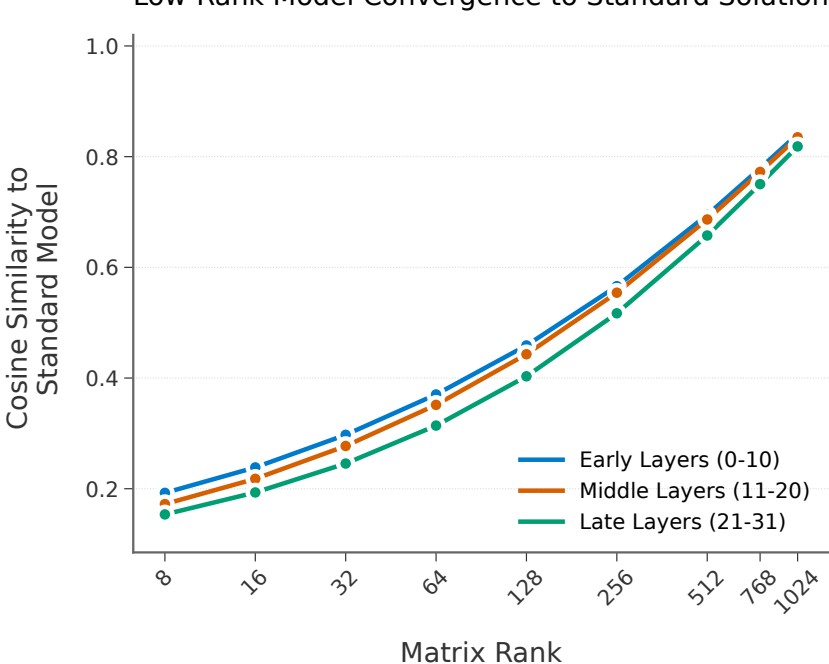

Figure 10: Cosine similarity between weight matrices from a standard fine-tuned model and a Nested Subspace Network. The similarity increases with rank but never reaches 1.0, indicating that the low-rank constraint guides the NSN to a different, yet effective, local minimum in the loss landscape.

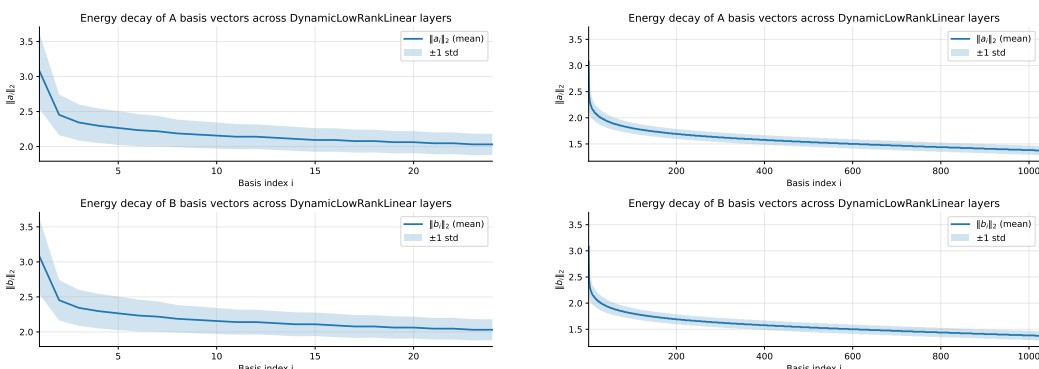

(a) **Short-horizon energy decay.** Mean and standard deviation of $\|\mathbf{a}_i\|_2$ and $\|\mathbf{b}_i\|_2$ over the first 24 basis components, showing the local decay structure across layers.

(b) **Full-range energy decay.** Decay profile across the entire rank spectrum, illustrating the global monotonic trend implied by Assumption 1.

Figure 11: **Empirical energy decay patterns across NSN layers.** Each plot aggregates the norms of rank-1 components $(\mathbf{a}_i, \mathbf{b}_i)$ across all DynamicLowRankLinear layers in the NSN-modified GPT-NeoX model. The short-range plot highlights early-index behavior, while the full-range plot captures the complete structural decay. Both views provide complementary evidence supporting the energy-ordering behaviour predicted by Assumption 1.

holds, we interpret them as noise in the optimization process. them as minor fluctuations introduced by the optimization dynamics (Table 4).

| Layer | R | A-mon. | A-viol. | B-mon. | B-viol. | $A_{\text{first}}$ | $A_{\text{last}}$ | $B_{\text{first}}$ |
|---|---|---|---|---|---|---|---|---|
| L0 h→4h | 1024 | F | 1 | F | 2 | 3.141 | 1.406 | 3.141 |
| L0 4h→h | 1024 | F | 2 | T | 0 | 2.578 | 1.188 | 2.562 |
| L1 h→4h | 1024 | T | 0 | T | 0 | 3.922 | 1.297 | 3.922 |
| L1 4h→h | 1024 | T | 0 | T | 0 | 2.797 | 1.195 | 2.781 |
| L2 h→4h | 1024 | T | 0 | F | 1 | 3.828 | 1.336 | 3.828 |
| L2 4h→h | 1024 | T | 0 | T | 0 | 2.969 | 1.258 | 2.953 |
| L3 h→4h | 1024 | F | 1 | F | 2 | 3.562 | 1.375 | 3.547 |
| L3 4h→h | 1024 | T | 0 | T | 0 | 2.578 | 1.234 | 2.578 |
| L4 h→4h | 1024 | T | 0 | T | 0 | 3.797 | 1.375 | 3.797 |
| L4 4h→h | 1024 | T | 0 | F | 1 | 2.938 | 1.227 | 2.938 |
| L5 h→4h | 1024 | F | 1 | T | 0 | 3.562 | 1.375 | 3.562 |
| L5 4h→h | 1024 | T | 0 | F | 1 | 2.609 | 1.266 | 2.609 |
| L6 h→4h | 1024 | F | 2 | F | 1 | 3.625 | 1.352 | 3.625 |
| L6 4h→h | 1024 | T | 0 | T | 0 | 2.531 | 1.281 | 2.531 |
| L7 h→4h | 1024 | T | 0 | F | 1 | 3.672 | 1.344 | 3.656 |
| L7 4h→h | 1024 | T | 0 | T | 0 | 2.641 | 1.281 | 2.641 |
| L8 h→4h | 1024 | F | 1 | T | 0 | 3.641 | 1.344 | 3.641 |
| L8 4h→h | 1024 | T | 0 | T | 0 | 2.531 | 1.281 | 2.531 |
| L9 h→4h | 1024 | F | 1 | F | 3 | 3.625 | 1.336 | 3.625 |
| L9 4h→h | 1024 | F | 1 | T | 0 | 2.484 | 1.273 | 2.484 |
| L10 h→4h | 1024 | F | 1 | F | 4 | 3.562 | 1.336 | 3.562 |
| L10 4h→h | 1024 | F | 1 | T | 0 | 2.312 | 1.273 | 2.312 |
| L11 h→4h | 1024 | F | 1 | T | 0 | 3.516 | 1.336 | 3.516 |
| L11 4h→h | 1024 | T | 0 | T | 0 | 2.312 | 1.281 | 2.312 |
| L12 h→4h | 1024 | T | 0 | T | 0 | 3.625 | 1.336 | 3.625 |
| L12 4h→h | 1024 | T | 0 | F | 2 | 2.344 | 1.289 | 2.344 |
| L13 h→4h | 1024 | T | 0 | T | 0 | 3.641 | 1.344 | 3.641 |
| L13 4h→h | 1024 | T | 0 | F | 1 | 2.266 | 1.312 | 2.266 |
| L14 h→4h | 1024 | T | 0 | T | 0 | 3.656 | 1.344 | 3.656 |
| L14 4h→h | 1024 | T | 0 | T | 0 | 2.266 | 1.336 | 2.266 |
| L15 h→4h | 1024 | T | 0 | T | 0 | 3.656 | 1.344 | 3.656 |
| L15 4h→h | 1024 | F | 1 | T | 0 | 2.422 | 1.344 | 2.422 |
| L16 h→4h | 1024 | T | 0 | T | 0 | 3.672 | 1.352 | 3.672 |
| L16 4h→h | 1024 | T | 0 | F | 1 | 2.344 | 1.359 | 2.344 |
| L17 h→4h | 1024 | T | 0 | F | 1 | 3.688 | 1.352 | 3.688 |
| L17 4h→h | 1024 | T | 0 | F | 2 | 2.375 | 1.391 | 2.375 |
| L18 h→4h | 1024 | F | 1 | T | 0 | 3.641 | 1.359 | 3.641 |
| L18 4h→h | 1024 | F | 1 | T | 0 | 2.453 | 1.414 | 2.453 |
| L19 h→4h | 1024 | T | 0 | T | 0 | 3.594 | 1.367 | 3.594 |
| L19 4h→h | 1024 | T | 0 | T | 0 | 2.609 | 1.422 | 2.625 |
| L20 h→4h | 1024 | T | 0 | T | 0 | 3.516 | 1.383 | 3.516 |
| L20 4h→h | 1024 | F | 1 | F | 1 | 2.828 | 1.438 | 2.828 |
| L21 h→4h | 1024 | T | 0 | F | 1 | 3.438 | 1.391 | 3.438 |
| L21 4h→h | 1024 | T | 0 | F | 1 | 2.984 | 1.453 | 2.984 |
| L22 h→4h | 1024 | F | 1 | T | 0 | 3.422 | 1.406 | 3.422 |
| L22 4h→h | 1024 | T | 0 | F | 1 | 3.047 | 1.477 | 3.047 |
| L23 h→4h | 1024 | F | 1 | T | 0 | 3.375 | 1.422 | 3.375 |
| L23 4h→h | 1024 | T | 0 | F | 1 | 2.734 | 1.500 | 2.750 |
| L24 h→4h | 1024 | T | 0 | T | 0 | 3.344 | 1.430 | 3.344 |
| L24 4h→h | 1024 | T | 0 | F | 2 | 2.484 | 1.516 | 2.484 |
| L25 h→4h | 1024 | T | 0 | F | 1 | 3.344 | 1.438 | 3.344 |
| L25 4h→h | 1024 | T | 0 | T | 0 | 2.328 | 1.523 | 2.328 |
| L26 h→4h | 1024 | F | 1 | T | 0 | 3.312 | 1.438 | 3.312 |
| L26 4h→h | 1024 | T | 0 | F | 1 | 2.234 | 1.531 | 2.234 |
| L27 h→4h | 1024 | T | 0 | T | 0 | 3.281 | 1.438 | 3.281 |
| L28 h→4h | 1024 | T | 0 | T | 0 | 3.281 | 1.438 | 3.266 |
| L28 4h→h | 1024 | F | 1 | T | 0 | 2.391 | 1.555 | 2.391 |
| L29 h→4h | 1024 | F | 1 | T | 0 | 3.266 | 1.438 | 3.266 |
| L29 4h→h | 1024 | T | 0 | T | 0 | 2.625 | 1.555 | 2.625 |
| L30 h→4h | 1024 | T | 0 | T | 0 | 3.297 | 1.438 | 3.297 |
| L30 4h→h | 1024 | T | 0 | T | 0 | 3.688 | 1.531 | 3.688 |
| L31 h→4h | 1024 | T | 0 | T | 0 | 3.562 | 1.422 | 3.562 |
| L31 4h→h | 1024 | T | 0 | T | 0 | 3.688 | 1.445 | 3.688 |

Table 4: **Empirical evaluation of Assumption 1 across all DynamicLowRankLinear layers.** For each layer, we compute the norms of the rank-1 components $(\mathbf{a}_i, \mathbf{b}_i)$. Columns indicate: maximum rank $R$, monotonicity flags for $A$ and $B$ (A-mon., B-mon.), the number of violations (A-viol., B-viol.), and the norms of the first and last components, which capture the magnitude of decay. This table quantifies how consistently the trained NSN architecture orders its basis directions by importance.

## C.5 ENERGY PROFILES IN STANDARD DENSE MODELS

To complement the analysis in Appendix C.4, we perform the same diagnostic procedure on a standard, unmodified GPT-NeoX model whose MLP blocks use conventional dense linear transformations. This comparison isolates whether the energy decay structure observed in NSNs also appears in ordinary architectures, or whether it is instead a property induced by the NSN reparameterization and training objective.

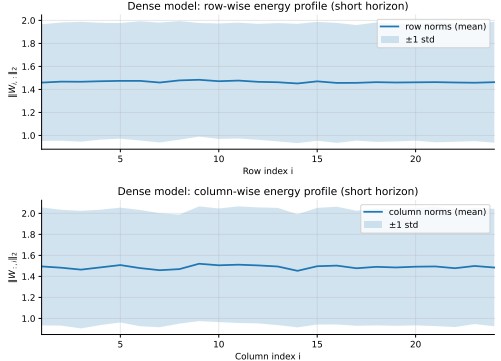 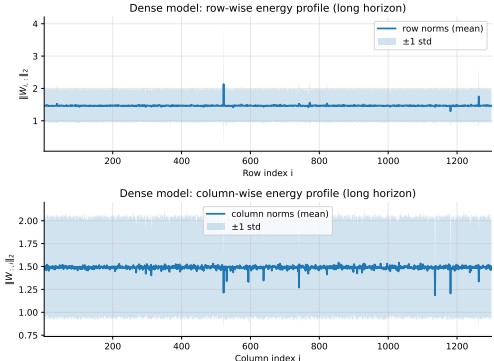

(a) **Short-horizon energy profile in a dense model.** Mean and standard deviation of row and column norms over the first 24 indices for the dense GPT-NeoX MLP layers. The profiles are essentially flat, indicating no systematic dependence of weight energy on index; all weights have comparable energy.

(b) **Full-range energy profile in a dense model.** Row- and column-wise norms across the entire index range show no clear trend or decay, consistent with an absence of any ordered structure in the weight energies. All indices exhibit similar magnitude, confirming that there is effectively no relationship between index and energy.

Figure 12: **Lack of ordered energy decay in standard dense GPT-NeoX MLP layers.** Unlike the NSN-adapted model, where rank-1 components exhibit a clear energy decay with basis index, the dense model's row and column norms are nearly constant across indices. This indicates that the dense parameterization does not naturally impose an ordering of directions by energy: all weights have effectively the same energy, and there is no meaningful relationship between index and importance.

**Setup**. For each dense MLP layer, we take the weight matrix $W \in \mathbb{R}^{d_{\text{out}} \times d_{\text{in}}}$ and examine its rows and columns directly, without any factorization. We define

$$\mathbf{a}_i = W_{i,:}, \qquad \mathbf{b}_i = W_{:,i},$$

and compute the Euclidean norms $\|\mathbf{a}_i\|_2$ and $\|\mathbf{b}_i\|_2$ across all row and column indices. This is the direct analogue of the NSN analysis: if dense models naturally encode more important directions earlier in their parameterization, we would observe structured energy decay across indices. We evaluate monotonicity, quantify violations, and compute layer-averaged energy profiles exactly as in the NSN case. The resulting aggregated profiles are visualized in Figures 12a and 12b.

**Findings**. Across all layers, the energy profiles are essentially flat: both row norms and column norms remain nearly constant as a function of index. Unlike the NSN-adapted model, where low-index basis vectors consistently exhibit higher energy and a clear decay pattern, the dense model displays no meaningful ordering. Monotonicity is neither present nor expected; the norms fluctuate minimally and show no global trend. This indicates that in standard dense architectures, parameter indices do not correspond to any notion of directional importance, and no analogue of Assumption 1 emerges from training alone.

**Takeaway**. The absence of any structured energy decay in dense models highlights a key distinction between NSNs and conventional architectures. Whereas NSNs learn a highly organized hierarchy of basis directions—with most of the functional energy concentrated in early rank components—dense models distribute energy uniformly with no discernible ordering. This comparison reinforces that the nested subspace structure arises from the NSN parameterization and training procedure rather than from generic properties of large neural networks. Figures 12a and 12b make this contrast explicit: NSNs exhibit sharp, consistent energy decay, whereas dense models show no relationship between index and energy at all.

**Violation Analysis Setup.** To quantify how strongly the dense model violates the energy–ordering property, we compute violation rates in direct analogy to the NSN analysis. For each MLP layer, we examine the sequences of row norms and column norms,

$$\mathbf{a}_i = W_{i,:}, \qquad \mathbf{b}_i = W_{:,i},$$

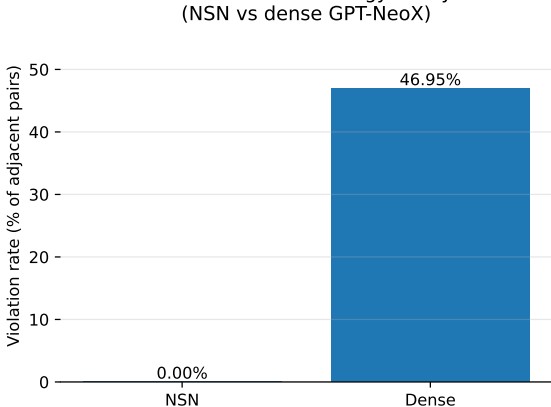

Figure 13: **Aggregate violation rate of the energy–decay assumption in NSN and dense models.** Bars show the percentage of adjacent index pairs that violate the monotonic decay condition, aggregated across all MLP layers. The NSN model exhibits extremely low violation rates (0.00% in this specific run), whereas the dense model shows violation rates that are similar to random orderings (which would be about 50%).

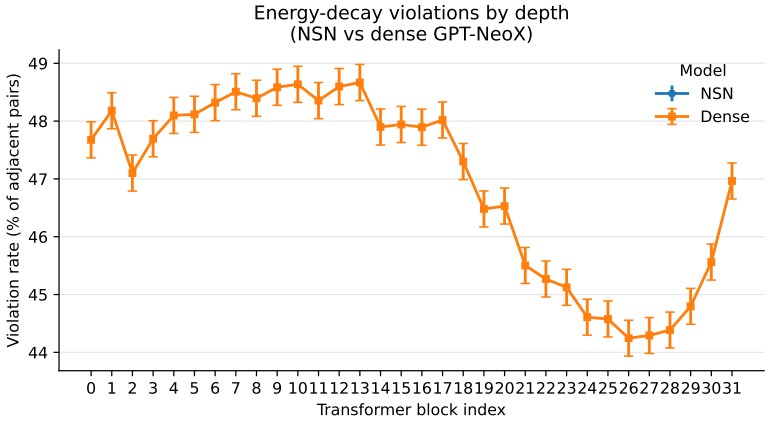

Figure 14: **Violation rate by transformer block.** Violation rates are shown per transformer block for both NSN (circles) and dense (squares) models, with binomial standard errors. The NSN error rates are not visible because they are negligible (below 0.01% on the graph). The dense model seems to have a consistent pattern how often the energy decay assumption is violated, yet this violation is consistently extremely high. This violation makes sense given that nearby transformer blocks should be correlated. The key takeaway is that regular training schemes do not induce a sufficient ordering of basis vectors.

and record how often adjacent pairs fail the monotonic condition $\|\mathbf{a}_i\|_2 \geq \|\mathbf{a}_{i+1}\|_2$ or $\|\mathbf{b}_i\|_2 \geq \|\mathbf{b}_{i+1}\|_2$. Each adjacent index yields a binary event (violation or no violation), allowing us to compute per-layer and per-model violation rates. For a fair comparison to NSNs, we aggregate all adjacent comparisons across all layers and report both an overall violation rate and a depth-resolved profile with binomial standard errors.

**Takeaway.** The violation statistics provide a complementary perspective to the energy profiles reported earlier. Across all layers and transformer blocks, NSNs demonstrate strikingly consistent adherence to the energy–decay structure, with only a tiny fraction of adjacent pairs (typically well below 1%) violating monotonicity. Dense models, by contrast, display no such structure: violation rates are an order of magnitude larger and show no systematic dependence on depth. Together, these

results reinforce that the nested subspace ordering is not an incidental artifact of large neural networks but a direct consequence of the NSN parameterization and training objective, which actively induce a stable and ordered hierarchy of basis directions.

## C.6 COMPUTATIONAL EFFICIENCY THROUGH SURGICAL REPLACEMENT

**Objective.** This analysis demonstrates the practical computational benefits of surgically replacing standard linear layers with rank-adaptive linear layers in existing transformer architectures. The goal is to quantify the reduction in floating-point operations (FLOPs) achieved through low-rank decomposition while maintaining the nested subspace structure.

**Methodology.** We performed surgical modifications to the Gemma-2B architecture by replacing all linear layers within the MLP blocks with rank-adaptive variants. Each original weight matrix $W$ was decomposed using Singular Value Decomposition (SVD) to initialize the factorized form $W \approx BA$, where $B$ and $A$ are lower-rank matrices. The MLP blocks, which contain three linear layers with GeLU activation functions, were systematically converted to support multiple rank configurations while preserving the original model's functionality.

**Results and Interpretation.** The surgical replacement approach enables significant computational savings through reduced matrix operations. By decomposing the original full-rank weight matrices into their low-rank approximations, the number of parameters and corresponding FLOPs are substantially reduced. This modification allows for dynamic rank selection during inference, providing a trade-off between computational efficiency and model capacity without requiring complete retraining of the base model.

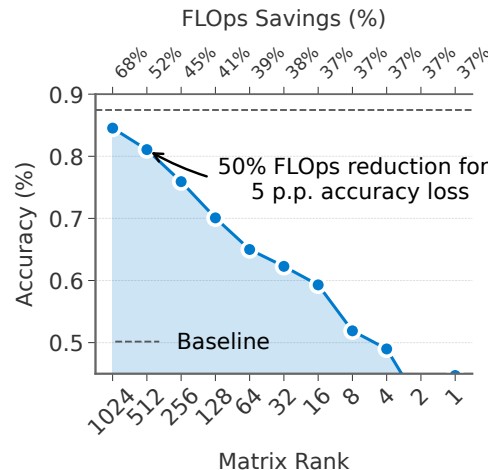

Figure 15: **Example Surgical Changes to linear layers only on Gemma-2B**. The architecture of Gemma-2B. All MLP blocks contain three linear layers with a GeLU activation function. We surgically replace all linear layers with *rank-adaptive* linear layers, initialized $W \approx BA$ via SVD-decomposition

# D ADDITIONAL THEORETICAL INSIGHTS ON NESTED SUBSPACE NETWORKS

## D.1 CONDITION FOR NESTED SUBSPACE PROPERTY

**Lemma 1** (Sufficient condition for nested subspace property). *If $rank(A_r) = r$ for all $r \leq R$, then $Im(W_r) = span\{b_1, \ldots, b_r\}$ and therefore $Im(W_r) \subseteq Im(W_{r+1})$.*

The SVD initialization as well as our rank training objective make rank deficiency extremely rare in practice.

## D.2 SIMPLE EXAMPLE OF AN NSN LAYER

**Example 1 (Toy NSN layer).** To make this construction concrete, consider an NSN layer with $d_{\text{in}} = d_{\text{out}} = 2$ and maximum rank $R = 2$. We choose a single pair of factor matrices

$$A = \begin{bmatrix} 1 & 0 \\ 0 & 1 \end{bmatrix}, \qquad B = \begin{bmatrix} 1 & 0 \\ 0 & 1 \end{bmatrix}.$$

For rank $r = 1$, we use only the first row of $A$ and the first column of $B$:

$$A_1 = \begin{bmatrix} 1 & 0 \end{bmatrix}, \qquad B_1 = \begin{bmatrix} 1 \\ 0 \end{bmatrix}, \qquad W_1 = B_1 A_1 = \begin{bmatrix} 1 & 0 \\ 0 & 0 \end{bmatrix}.$$

For rank $r = 2$, we use all rows/columns:

$$A_2 = A, \qquad B_2 = B, \qquad W_2 = B_2 A_2 = \begin{bmatrix} 1 & 0 \\ 0 & 1 \end{bmatrix}.$$

Thus, the rank-1 and rank-2 effective weights $W_1$ and $W_2$ are both derived from the same factor matrices $(A, B)$. Adjusting the rank simply changes how many basis vectors are active.

## D.3 WHY USE THE UNCERTAINTY-AWARE OBJECTIVE?

NSNs train a hierarchy of rank-truncated submodels inside one set of weights by factorizing each linear layer $W = BA$ and enforcing a nested-subspace structure across ranks. We view training across ranks as a multi-task learning problem and propose to use a Kendall-style objective for this. Empirically, the learned log-variances decrease with rank which we interpret as a proxy for rank expresiveness. In our ablations, we show that two cross-entropies deliver the required gains across ranks and adding additional regularization is not productive.

**What properties do we look for?** We seek a weighting mechanism that (i) automatically adapts the relative importance of ranks without per-rank hyperparameter tuning, (ii) is invariant to arbitrary rescalings of the factorization $W = BA$, (iii) guarantees positive weights, and (iv) is cheap enough to apply inside every NSN layer and on every training step. The uncertainty-aware objective surrogate satisfies these requirements: the reparameterization $\exp(-s_k)$ yields strictly positive, smoothly varying weights with a strictly convex dependence on $s_k$, so optimization is stable, and because it operates on loss values rather than gradient norms it is insensitive to the scale ambiguity between $A$ and $B$.

**What are the benefits of using an uncertainty-aware objective?** The uncertainty-aware objective directly addresses the heterogeneous difficulty of learning different ranks: lower ranks exhibit larger and noisier cross-entropy losses, which would otherwise dominate or destabilize the optimization if all ranks were weighted equally. Introducing rank-specific log-variances $s_k$ yields effective weights $\exp(-s_k)$ that adaptively attenuate gradients from high-uncertainty ranks while the additive $s_k$ term prevents trivial suppression of a task, so the hierarchy is trained jointly but stably in a single objective. This surrogate empirically leads to well-behaved performance at low ranks and at interpolated ranks that were never explicitly optimized. In addition, the learned $s_k$ form an interpretable diagnostic: higher ranks consistently converge to lower log-variances than lower ranks (Fig. 3), providing a quantitative measure of rank expressiveness rather than treating the rank index as a purely architectural hyperparameter.

**Would we use a different mechanism for weighting the ranks?** In principle, any multi-task reweighting scheme could be applied across ranks, even methods such as GradNorm (Chen et al.,

2018), but these alternatives come with trade-offs that are poorly matched to the NSN parameterization. For instance, gradient-norm–based methods require choosing a reference layer and repeatedly computing per-rank norms, and their behavior is sensitive to the arbitrary scaling between $A$ and $B$ in $W = BA$, so the induced weights can drift for reasons unrelated to rank difficulty. Moreover, aggressively equalizing training rates across ranks can over-emphasize very low ranks early in training and degrade the anchor model, whereas our anchor–variant design intentionally biases optimization toward a strong high-rank solution while still improving smaller ranks. For these reasons we view the uncertainty-aware objective as the most practical default for NSNs, and leave more elaborate, possibly model-specific weighting schemes as future work rather than as necessary components of the method.

### D.4 PROOF OF PROPOSITION

We seek to demonstrate that the performance of the network at an untrained, interpolated rank $r_{\text{int}}$ remains close to the performance at an explicitly trained rank. This property relies on the structure induced by the training process. Let the shared, learned weight matrices be $A \in \mathbb{R}^{R \times d_{\text{in}}}$ and $B \in \mathbb{R}^{d_{\text{out}} \times R}$, where $R$ is the maximum rank. Let $\mathbf{a}_i \in \mathbb{R}^{1 \times d_{\text{in}}}$ be the $i$-th row vector of $A$ and $\mathbf{b}_i \in \mathbb{R}^{d_{\text{out}} \times 1}$ be the $i$-th column vector of $B$.

**Lemma 2** (Adjacent Rank Perturbation). *Let $f(\mathbf{x}; r) = (\sum_{i=1}^{r} \mathbf{b}_i \mathbf{a}_i)\mathbf{x}$ be the output of the linear layer for an input $\mathbf{x}$ at rank $r$. The perturbation to the output when moving from rank $r$ to $r + 1$ is bounded by:*

$$\|f(\mathbf{x}; r + 1) - f(\mathbf{x}; r)\| \le \|\mathbf{b}_{r+1}\| \|\mathbf{a}_{r+1}\| \|\mathbf{x}\|$$

*Proof.* The change in the weight matrix is $W_{r+1} - W_r = \mathbf{b}_{r+1}\mathbf{a}_{r+1}$. The change in the output is thus $(W_{r+1} - W_r)\mathbf{x}$. Applying the submultiplicative property of matrix and vector norms yields the result: $\|(\mathbf{b}_{r+1}\mathbf{a}_{r+1})\mathbf{x}\| \le \|\mathbf{b}_{r+1}\mathbf{a}_{r+1}\| \|\mathbf{x}\| \le \|\mathbf{b}_{r+1}\| \|\mathbf{a}_{r+1}\| \|\mathbf{x}\|$. $\qquad\square$

*Proof of Proposition 1.* The total change in the function output between rank $r_1$ and $r_{\text{int}}$ can be expressed as a telescoping sum. By the triangle inequality and Lemma 2:

$$\|f(\mathbf{x}; r_{\text{int}}) - f(\mathbf{x}; r_1)\| = \left\| \sum_{i=r_1+1}^{r_{\text{int}}} (f(\mathbf{x}; i) - f(\mathbf{x}; i - 1)) \right\|$$

$$\le \sum_{i=r_1+1}^{r_{\text{int}}} \|f(\mathbf{x}; i) - f(\mathbf{x}; i - 1)\|$$

$$\le \left( \sum_{i=r_1+1}^{r_{\text{int}}} \|\mathbf{b}_i\| \|\mathbf{a}_i\| \right) \|\mathbf{x}\|$$

Due to the Lipschitz continuity of the loss $\mathcal{L}$, the difference in expected error is bounded:

$$|E(r_{\text{int}}) - E(r_1)| \le L_{\mathcal{L}} \cdot \mathbb{E}\left[ \|f(\mathbf{x}; r_{\text{int}}) - f(\mathbf{x}; r_1)\| \right]$$

Substituting the bound on the function perturbation and defining $C = L_{\mathcal{L}} \cdot \mathbb{E}[\|\mathbf{x}\|]$ yields the final result. $\qquad\square$

### D.5 SVD INITIALIZATION

We propose an initialization strategy based on Singular Value Decomposition that preserves the original model parameterization at the outset of training. Formally, for a given pre-trained weight matrix $W$, we compute its SVD, $W = U\Sigma V^T$, and initialize the factor matrices $B \in \mathbb{R}^{d_{out} \times \tilde{R}}$ and $A \in \mathbb{R}^{\tilde{R} \times d_{in}}$ using the top $\tilde{R}$ singular components: $B_{\text{init}} := U_{\tilde{R}} \sqrt{\Sigma_{\tilde{R}}}$ and $A_{\text{init}} := \sqrt{\Sigma_{\tilde{R}}} V_{\tilde{R}}^T$, where $U_{\tilde{R}}$ and $V_{\tilde{R}}$ contain the first $\tilde{R}$ columns of $U$ and $V$, and $\Sigma_{\tilde{R}}$ is the diagonal matrix of the top $\tilde{R}$ singular values. This scheme ensures that at the maximum rank $\tilde{R}$, the NSN layer's effective weight matrix, $W_{\tilde{R}} = B_{\text{init}}A_{\text{init}}$, reconstructs the original pre-trained matrix $W$ either exactly (if $\tilde{R} = R$) or with the smallest Frobenius norm (if $\tilde{R} < R$).

### D.6 TRAINING COST OF NSN

Compared to a standard dense network with weight matrix $W \in \mathbb{R}^{d_{\text{out}} \times d_{\text{in}}}$, whose per-step training cost is proportional to $d_{\text{in}} d_{\text{out}}$ FLOPs, training a Nested Subspace Network (NSN) layer with maximum training rank $\tilde{R}$ and a sampled variant rank $r < \tilde{R}$ requires per-step FLOPs proportional to $(\tilde{R} + r)(d_{\text{in}} + d_{\text{out}})$, because each optimization step performs a forward–backward pass at the anchor rank $\tilde{R}$ and another at the variant rank $r$. Using the break-even rank $R_{\text{be}} = \frac{d_{\text{in}} d_{\text{out}}}{d_{\text{in}} + d_{\text{out}}}$, for which a single low-rank pass matches the dense cost, the pessimistic case $r \approx \tilde{R} \approx R_{\text{be}}$ gives a total of about $2 d_{\text{in}} d_{\text{out}}$ FLOPs per step, i.e., at most roughly twice the cost of training one dense model with the same input and output dimensions. However, this single NSN training run yields a whole hierarchy of usable ranks at test time, so if $K$ different computational budgets are needed, the NSN still replaces $K$ separate dense training runs (total cost $\approx K d_{\text{in}} d_{\text{out}}$) with one run whose cost is only a constant factor above that of a single dense model, effectively amortizing the training cost over many operating points.

### D.7 ON THE DERIVED FUNCTIONAL FORM OF THE LOSS IN EQUATION

Why did we arrive at the specific functional form in Equation 3 and, concretely, why are we using the exponential as the coefficient?

We start with our goal: We want a positive weight for each rank–specific loss that adapts during training but does not collapse to zero or infinity. Let $L_k$ denote the task loss at rank $k$ and define a positive weight $w_k > 0$. We can write out two equivalent parametrizations which are useful in our context:

**1) Direct optimization over positive weights with a log–barrier**

$$\min_{w_k > 0} \sum_k \left[ w_k L_k \ - \ \log w_k \right].$$

The term $-\log w_k$ prevents $w_k \to 0$ and yields a unique closed–form optimum in $w_k$ for fixed model parameters. We can reparametrize this equation to yield an equivalent re-parametrization found in the main body of the paper.

**2) Reparameterizing $w_k = e^{-s_k}$ with $s_k \in \mathbb{R}$**

$$\sum_k \left[ e^{-s_k} L_k + s_k \right].$$

This matches Eq. 3 (up to a constant), with $s_k = \log \sigma_k^2$.

Why this particular form of the optimization? are a few different ways to think about it.

**(a) Positivity and simple optimization**  The mapping $w_k = e^{-s_k}$ guarantees $w_k > 0$ for all $s_k$. Furthermore, the objective is convex in $s_k$ for fixed $L_k$:

$$\frac{\partial^2}{\partial s_k^2} \left( e^{-s_k} L_k + s_k \right) = e^{-s_k} L_k > 0.$$

If $L_k > 0$, then $\frac{\partial^2}{\partial s_k^2} \left( e^{-s_k} L_k + s_k \right) > 0$ for all $s$. This means the function is stricly convex in $s$. If $L_k = 0$, then the loss is $s_k$ which is still convex. This convexity is a useful property for gradient updates and helps to learn the different contributions effectively, as empirically shown in Fig. 3.

**(b) Closed–form optimal weights and scale invariance**  This parametrization allows to obtain easy closed-form weights. For fixed model parameters,

$$\frac{\partial}{\partial s_k} \left( e^{-s_k} L_k + s_k \right) = -e^{-s_k} L_k + 1.$$

Setting this to zero gives

$$e^{-s_k^\star} = \frac{1}{L_k}, \qquad w_k^\star = \frac{1}{L_k}.$$

This gives us two useful properties:

- **Loss–scale invariance:** If $L_k \leftarrow cL_k$, then $s_k^\star \leftarrow s_k^\star + \log c$ while $w_k L_k = 1$ remains unchanged.

- **Coarse gradient balancing:** At the optimum, the contribution of rank $k$ is

$$w_k \nabla_\theta L_k = \frac{1}{L_k} \nabla_\theta L_k,$$

which prevents dominance by a loss with artificially large scale. This is a particularly useful property since we *expect* models with lower ranks to have higher loss due to their (definitionally) lower expressivity. Recall that Eq. 3 scales gradients by $\nabla_\theta L_{\text{total}} = \sum_k e^{-s_k} \nabla_\theta L_k$. Therefore, jointly learning $s_k$ allows the optimizer to attenuate gradients from noisier ranks ($s_k$ large) and amplify gradients from cleaner ranks ($s_k$ small), while the term $+s_k$ prevents collapse $e^{-s_k} \to 0$.

**(c) Link to heteroskedastic uncertainty** We can think of this loss as being directly tied to heteroskedsatic uncertainty in the regression case. Concretely, for Gaussian regression noise, the negative log–likelihood is

$$\frac{1}{2\sigma^2} \|\text{residual}\|^2 + \frac{1}{2} \log \sigma^2.$$

Setting $s_k = \log \sigma^2$ gives the structure $e^{-s_k}(\cdot) + s_k$. Classification lacks a Gaussian residual. However, it is common in practice to use this as a surrogate objective. Equation 3 acts as a surrogate for such a Gaussian residual in the classification setting.

## D.8 Why log-variances are emergent proxies for expresiveness of each rank

**Are the log-variances free parameters?** The log-variances are not free parameters, but they are trainable parameters. They are not free because in the objective

$$e^{-s_k} \mathcal{L}_{\text{CE}}(k) + s_k,$$

each log-variance $s_k$ is coupled to the rank-$k$ loss. This coupling means their values depend directly on the loss within each model of a given rank. However, we still learn these values during training.

**Why does log-variance serve as an emergent proxy? Short answer:** This parameter tracks the residual loss for each rank. Higher residual loss (higher error for a given rank) leads to a higher learned uncertainty parameter. Thus, it emerges as a proxy for expressiveness: higher residual loss indicates a less expressive model, and the parameter tracks this loss.

Each rank-model contributes differently to the training objective. For a rank $k$ model, the contribution is

$$\mathcal{L}_k = e^{-s_k} L_{\text{CE}}(k) + s_k,$$

where $s_k$ is the log-variance and $L_{\text{CE}}(k)$ is the cross-entropy loss.

After training, at a stationary point where the gradient is zero, we have

$$\frac{\partial \mathcal{L}_k}{\partial s_k} = -e^{-s_k} L_{\text{CE}}(k) + 1 = 0,$$

which implies

$$e^{-s_k} L_{\text{CE}}(k) = 1,$$

and therefore

$$s_k = \log L_{\text{CE}}(k).$$

Thus, up to optimization noise and interactions with other parameters, the learned log-variances track the scale of the residual loss at that rank.

**More expressive vs. less expressive models.** A more expressive model can reduce $L_{\text{CE}}(k)$ further, which forces $s_k$ to be smaller. A less expressive model is stuck with a higher $L_{\text{CE}}(k)$ and therefore learns a higher $s_k$. Over training, this creates a relationship in which ranks that explain the data well end up with lower log-variances, while ranks that explain it poorly end up with higher log-variances.

---

**Algorithm 2** Forward pass of a Nested Subspace Network (NSN) layer

---

**Require:** Input vector batch $X \in \mathbb{R}^{B \times d_{\text{in}}}$
**Require:** Factor matrices $A \in \mathbb{R}^{R \times d_{\text{in}}}$, $B \in \mathbb{R}^{d_{\text{out}} \times R}$, bias $b \in \mathbb{R}^{d_{\text{out}}}$
**Require:** Active rank $r \in \{1, \ldots, R\}$
**Ensure:** Output logits $Y \in \mathbb{R}^{B \times d_{\text{out}}}$
  1: $A_r \leftarrow$ first $r$ rows of $A$                                                     $\triangleright A_r \in \mathbb{R}^{r \times d_{\text{in}}}$
  2: $B_r \leftarrow$ first $r$ columns of $B$                                       $\triangleright B_r \in \mathbb{R}^{d_{\text{out}} \times r}$
  3: $H \leftarrow X A_r^\top$                   $\triangleright$ Project inputs to rank-$r$ subspace, $H \in \mathbb{R}^{B \times r}$
  4: $Y \leftarrow H B_r^\top + \mathbf{1} b^\top$                          $\triangleright$ Map back to output space
  5: **return** $Y$

---

**Algorithm 3** Forward pass of a Nested Subspace Network

---

**Require:** Input batch $X$
**Require:** NSN layers $\{\text{Layer}_\ell\}_{\ell=1}^L$, each with $(A_\ell, B_\ell, b_\ell)$ and shared max rank $R$
**Require:** Active rank $r \in \{1, \ldots, R\}$
**Ensure:** Output logits $Z$
  1: $H \leftarrow X$
  2: **for** $\ell = 1$ to $L$ **do**
  3:     $H \leftarrow \text{NSNLayerForward}(H, A_\ell, B_\ell, b_\ell, r)$                 $\triangleright$ Alg. 2
  4:     **if** $\ell < L$ **then**
  5:         $H \leftarrow \phi(H)$             $\triangleright$ Apply nonlinearity, e.g. ReLU or GELU
  6:     **end if**
  7: **end for**
  8: $Z \leftarrow H$                                                $\triangleright$ Final logits
  9: **return** $Z$

---

# E  PSEUDOCODE

---

**Algorithm 4** Multi-rank uncertainty-weighted training for Nested Subspace Networks (anchor at maximal rank)

---

**Require:** Training dataset $\mathcal{D} = \{(x_i, y_i)\}$
**Require:** Maximal rank $R$ and set of trainable ranks $\mathcal{K} \subseteq \{1, \ldots, R\}$
**Require:** NSN model with parameters $\theta = \{A_\ell, B_\ell, b_\ell\}_{\ell=1}^{L}$
**Require:** Rank-specific log-variances $\{s_k\}_{k \in \mathcal{K}}$ with $s_k = \log(\sigma_k^2)$
**Require:** Optimizer Opt
**Ensure:** Trained NSN parameters $\theta$ and log-variances $\{s_k\}$
 1: Initialize $\theta$ and set $s_k \leftarrow 0$ for all $k \in \mathcal{K}$
 2: **for** each training step **do**
 3:    Sample a minibatch $(X, Y) \sim \mathcal{D}$
 4:    $\tilde{R} \leftarrow R$                               ▷ Anchor rank is always the maximal rank
 5:    $\mathcal{K}_{\text{var}} \leftarrow \{k \in \mathcal{K} : k < \tilde{R}\}$
 6:    $r \leftarrow \text{UniformSample}(\mathcal{K}_{\text{var}})$       ▷ Variant rank is sampled from lower trainable ranks
 7:    Set model rank $r_{\text{active}} \leftarrow \tilde{R}$
 8:    $Z_{\tilde{R}} \leftarrow \text{NSNForward}(X, r_{\text{active}})$
 9:    $\mathcal{L}_{\text{CE}}(\tilde{R}) \leftarrow \text{CrossEntropy}(Z_{\tilde{R}}, Y)$
10:    Set model rank $r_{\text{active}} \leftarrow r$
11:    $Z_r \leftarrow \text{NSNForward}(X, r_{\text{active}})$
12:    $\mathcal{L}_{\text{CE}}(r) \leftarrow \text{CrossEntropy}(Z_r, Y)$
13:    $s_{\tilde{R}} \leftarrow$ log-variance associated with rank $\tilde{R}$
14:    $s_r \leftarrow$ log-variance associated with rank $r$
15:    $\mathcal{L}_{\text{anchor}} \leftarrow \exp(-s_{\tilde{R}})\,\mathcal{L}_{\text{CE}}(\tilde{R}) + s_{\tilde{R}}$
16:    $\mathcal{L}_{\text{variant}} \leftarrow \exp(-s_r)\,\mathcal{L}_{\text{CE}}(r) + s_r$
17:    $\mathcal{L}_{\text{total}} \leftarrow \mathcal{L}_{\text{anchor}} + \mathcal{L}_{\text{variant}}$
18:    Opt.zero_grad()
19:    Backpropagate gradients of $\mathcal{L}_{\text{total}}$ with respect to $\theta$ and $\{s_k\}$
20:    Opt.step()
21: **end for**

---

# F ON LLM USAGE

The authors have used large language models for three purposes:

- We have used LLMs to aid or polish our writing. This includes rephrasing text, shorterning, proof-reading for ambuigities or finding mistakes or inconsistencies in notation

- We used LLMs as a supplementary source of finding related work. While we have primarily performed related work searches via google scholar, we have used the "Deep Research" functionality to find other related work that we might have missed. This has resulted in us adding response-based KD and self-distillation as related work to the paper.

- We have used LLMs for research ideation early on in the paper. This included brainstorming ways how to make efficient deep neural networks, what are the properties that such neural networks should have, among others.

Otherwise, all the ideas presented in the paper are our own. We take full responsibility for any errors found in the paper.

–

