# OpenReview forum: "Deep Hierarchical Learning with Nested Subspace Networks for Large Language Models"
_ICLR.cc/2026/Conference — ICLR 2026 Poster_

### Official Review · Reviewer_TDZV · 2025-10-29

**Soundness:** 2
**Presentation:** 2
**Contribution:** 2
**Rating:** 4
**Confidence:** 3

**Summary:**

This work proposes a way to re-parametrize linear layers into low-rank factorization $W=A^{R \times d_{in}} \cdot B^{d_{out} \times R}$ such that the compute budget can be dynamically adjusted without retraining by constructing the  effective weight matrix from the first r rows of A and first r columns of B. In other words, the effective layers forms a filter chain of images for the ranks r < R (a.k.a Nested Subspace Property / Layer). To train the layer effectively, the work leverages an uncertainty-weighted training objective by Kendall+2018 that balances the learning across ranks. The experiments show that the nested subspace layers can be introduced into pretrained LLMs to create a smooth rank-based accuracy-vs-compute trade-off that remains adjustable at test-time.

**Strengths:**

- The approach is highly flexible and practical as it can be broadly applied to any pretrained models with linear layers through a straight forward Singular Value Decomposition. Rather than requiring the specification of a suitable rank up front that may not be known for a given problem and architecture the method allows to "shut up and train" to find an appropriate effective rank.
- The resulting architecture is intepretable in the sense that it is straight-forward to inspect how much each ranks learns to represent both empirically and theorectically (the offered theoretical discussion provides a good intution of the approximation trade-offs involved).

**Weaknesses:**

- Section 4.1 investigates whether a single NSN, trained with the multi-rank objective, can match the performance of multiple, specialized model, but only does so for an somewhat artificial image classification setup where regular MLP baselines are trained from scratch. What is currently missing is a similiar quantitive comparison with LoRA baselines to establish whether an NSN evaluated for rank r matches a LoRA model that exclusively trained for rank r. This would make this work comparable with prior baselines and delinate the benifits of NSN learning vs predefined-rank models.
- An explicit desideratum is granularity of adaption, but the trade-off here only works in discrete 'matrix rank' steps for each layer. Since different layers at different depths learn to represent different things it is not clear that a uniform rank pick across the entire model is ideal. Arguably, the most granular adaption would be picking the ideal rank size for each layer but it is not obvious how to do that efficiently.
- The work mentions the connection to alternative weight-learning approaches such as GradNorm but there is no comparison or analysis on why the Kendall-loss formulation is the best choice.

**Additional comments**

- The core idea of the work could be introduced and visualized more clearly, e.g. by illustrating an example A and B and the resulting effective weight matrix. The right column of Figure 1 does not make it very clear that there is only 1 instance of A and B and not a duplication of many A and Bs.
- L182 "Following Kendall et al. (2018), we use the standard uncertainty-weighted surrogate objective"; I recommend expanding on this to make the work more self-contained and accessible to readers who may not be familiar with
- "a strict and dynamic computational budget" - I found this confusing to read; it might be better to concretely lay out the use cases that this work has in mind.
- L115 typo: that *form* a nested hierachy

Aside: The manuscript uses visually highlighted "Takeaway" messages that spell out what the reader may conclude from each section. I am not sure if this is helpful because it can feel repetitive, or, if it does not align with the reader's conclusion, somewhat presumptuous. For example, L276 "smooth and predictable trade-off" placed before any experimental results reads more like an unverifiable claim than a takeaway.

**Questions:**

- It is not clear how the anchor and variant rank are chosen. Section 2.2 says that the ranks are "sampled" but I am not sure why and how specifically. Is this applied consistently across training runs or is there a random element to it?
- It is claimed that the log-variances "serve as an emergent proxy for the effective expressiveness of each rank-specific model" but it is not clear why that needs to be the case since they are as far as I can tell freely trainable parameters. The same goes for the "uncertainty-weighted surrogate" - how do we know that these parameters act as a meaningful surrogate for uncertainty?
- Are you planning to release your source code and a drop-in layer implementation?

---

> ### Author Response · Authors · 2025-11-20
> **Response (1/4)**
>
> Dear reviewer TDZV,
>
> Thank you for taking the time to carefully review our work. We're glad you found it highly flexible, practical, and interpretable. We address your concerns below **(A) - (K)**.
>
> **Brief summary**. In the revision we **(i)** clarify the relationship to LoRA and other low‑rank adapters, explaining why LoRA’s rank does not control FLOPs and is therefore not a possible comparison on the same compute frontier, **(ii)** add analysis and discussion showing how NSNs induce a layer‑wise specialization pattern, and **(iii)** make the uncertainty‑weighted objective explicit, including its optimization properties and its advantages over gradient‑norm–based alternatives.
>
> We have also added a summary of the **15 experiments** we've conducted in the paper, as well as the **11 changes to the paper** after the rebuttal in the global response which we encourage you to take a look.
>
> ## (A) Comparing to a LoRA-tuned model
>
> Thank you for raising this point. This comparison seems natural at first glance but there's a nuanced reason why we did not do it in the paper (which we could have communicated better).
>
> The reason we did not compare to a LoRA-rank-r model is because *LoRA's rank does not control the computational budget*. Concretely, a LoRA adapter introduces a low-rank *update* $W \leftarrow W + BA$, but the $W$ is frozen. So, this does not alter the FLOP cost or active parameter count of the underlying layer. In other words, LoRA models do not lie on a compute-performance frontier, so it's practically not possible to change their "FLOPs". In contrast, NSNs *do* change FLOPs as you change the rank because we do not freeze the weights $W$.
>
> To put it extremely simply: we developed this model to solve a practical problem we faced ourselves: we wanted a single model that could in fact have multiple "sub-models" within it, each sub-model being a "cheaper" model that can run faster  but is less expressive. We can now do this: when we pick a lower-rank model (e.g. rank-32), we do in fact get a cheaper/smaller model. This is *not* the case with LoRA because it does not reduce the size of the network.
>
> We revised the paper to clarify this by adding the following in the main paper (**L32-326**):
>
> > We focus on MLPs because they represent a natural benchmark for the expressive capacity for each parameter budget. We do not focus on LoRA adapters as a baseline because they do not alter the FLOP cost or active parameter count of a network
>
> We have also changed the introduction to better position what our work aims to do (**L32-38**).
>
> **Actions taken.** (a) Added clarification on scope of evaluation; (b) Improved intro positioning
>
> ---
>
> ## (B) Granularity of adaptation and picking the rank at each layer
>
> Thank you for your comment. We 100\% agree with you: we think, ideally, a network should shrink only in some layers and not others. It would be ideal, as you said, if each layer would adapt only in as much as it needs to. Our current formulation shrinks all layers up to rank-$r$. We think your proposal to make layer-specific shrinkage is a non-trival extension that requires significant experimentation. We therefore look at this as extremely fruitful future work that we hope our work inspires.
>
> To make this more concrete, we've added this exact point in our discussion paper (**L535-539**):
>
> > Currently, NSNs shrink or augment all layers to the same rank; we think an interesting--and nontrivial-- future work is to develop layer-specific mechanisms for adaptive compute. This requires, however, solving the difficult problem of correlating problem-specific information with layer-specific representational capacity, a problem that has so far attracted little attention.
>
> **Actions taken.** Expanded the discussion section on where future work can improve this modeling paradigm.

---

> ### Author Response · Authors · 2025-11-20
> **Response (2/4)**
>
> ## \(C) Why choose uncertainty-weighted loss formulation?
>
> There is, in fact, a principled reason why we chose the objective we did; we realize we did not include this reasoning in the paper. This is motivated by the properties our task has.
>
> **What properties do we look for?**
> Our task is very specific and has some key properties we look for. Specifically, we seek a weighting mechanism that **(i)** automatically adapts the relative importance of ranks without per-rank hyperparameter tuning, **(ii)** is invariant to arbitrary rescalings of the factorization $W = BA$, **(iii)** guarantees positive weights, and (iv) is cheap enough to apply inside every NSN layer and on every training step. The uncertainty-aware objective surrogate satisfies these requirements: the reparameterization $\exp(-s_k)$ yields strictly positive, smoothly varying weights with a strictly convex dependence on $s_k$, so optimization is stable, and because it operates on loss values rather than gradient norms it is insensitive to the scale ambiguity between $A$ and $B$. Specifically:
> - Lower ranks exhibit larger and noisier cross-entropy losses, which would otherwise dominate or destabilize the optimization if all ranks were weighted equally.
> - Introducing rank-specific log-variances $s_k$ yields effective weights $\exp(-s_k)$ that adaptively attenuate gradients from high-uncertainty rank.
>
> **Could we use a different mechanism (e.g. GradNorm)?**
>
> Perhaps, but these alternatives come with trade-offs that are poorly matched to the NSN parameterization. For instance, gradient-norm–based methods require choosing a reference layer and repeatedly computing per-rank norms, and their behavior is sensitive to the arbitrary scaling between $A$ and $B$ in $W = BA$, so the induced weights can drift for reasons unrelated to rank difficulty. Moreover, aggressively equalizing training rates across ranks can over-emphasize very low ranks early in training and degrade the anchor model, whereas our anchor–variant design intentionally biases optimization toward a strong high-rank solution while still improving smaller ranks.
>
> For these reasons we view the uncertainty-aware objective as the most practical default for NSNs, and leave more elaborate, possibly model-specific weighting schemes as future work rather than as necessary components of the method.
>
> We have made this clear in the paper by introducing multiple changes.
>
> **Actions taken** (a) We added "what properties should this optimization objective satisfy" in the main paper (**L183-190**); (b) We added why the derived equation is useful in practice in the main paper (**L228-235**); \(c) We add a full discussion why use the uncertainty-aware objective (**L1480 - 1503**); (d) We explain why other mechanisms do not immediately meet these properties (**L1504-1511**). (e) We explain why our optimization satisfies these properties in the main paper (**L240-243**).
>
> ---
>
> ## (D) Illustrating an example A and B, and the resulting effective weight matrix
>
> We see where the duplication of matrices $A$ and $B$ may have caused confusion, thank you for raising the point. We've addressed this in multiple places.
>
> (a) We have added an explanation of the matrix factorization after our definition (**L138-141**)
>
> > Note that there is a single pair of factor matrices $(A,B)$ for an NSN layer, and changing the rank $r$ only changes how many of their rows/columns are used to form $W_r$, not the underlying parameters.
>
> (b) We have updated the Figure caption of our model to better reflect this (**L69-L74**):
>
> > Nested Subspace Networks replace each linear layer with a \emph{single} pair of shared factor matrices $(A,B)$ defining a rank-trainable layer. The effective weight at rank $r$, $W_r = B_r A_r$, is obtained by using only the first $r$ rows of $A$ and first $r$ columns of $B$. Different operating points (different ranks) therefore correspond to using different prefixes of the same $(A,B)$ This allows for the construction of a compute-performance Pareto frontier at inference time
>
> \(c) We have added an intuitive, worked-out example how the matrices work **(L1460-1476)** and gave a reference in the main paper (**L140**).
>
> **Actions taken**. The three actions (a) - \(c) from above.
>
> ---
>
> ## (E) Making the uncertainty-weighted objective more self-contained
> Thank you for the recommendation. We've fix this in the paper, we think it looks better now.
>
> **Actions taken.** (a) Added an explicit algorithm (Algorithm 1) that describes the uncertianty objective clearly (**L216-226**); (b) Explained why the exponentials appear in the equation (**L228-235**); \(c) Explained the algorithm in the text (**L247-250**); (d) Added an appendix with the full pseudocode **(Appendix E)**

---

> ### Author Response · Authors · 2025-11-20
> **Response (3/4)**
>
> ## (F) Use cases of this work
>
> You mentioned that the words "strict and dynamic computational budget" were confusing. Based on your suggestion, we've re-written the first paragraph of the intro to include concrete use cases and make it more friendly to readers outside of this area. We've now revised it as follows:
>
> > **Motivation**. When we deploy deep learning-based systems in practice, there is a trade-off between two properties: how good the model is (*performance*) and how expensive it is to run (*compute*). Typically, the *larger* the model, the *better* the performance. When using such models at inference (deployment) time, we may want to choose, on-the-fly, how "expensive" vs "fast" a model should be. For instance, we may prefer (i) cheaper models for easier questions in language models; (ii) lower-compute models on phones when battery levels drop; or (iii) more expensive models for safety-critical requests such as medical diagnosis. In this paper, we consider exactly this problem---how to build a single network that can flexibly trade off performance and inference cost at test time.
>
> **Actions taken**. Updated first intro paragraph to position the work better.
>
> ---
>
> ## (G) Typo
>
> Fixed!
>
> ---
>
> ## (H) On the takeaways
>
> We understand the takeaway message felt repetative. We think it might be the case because we were highlighting similar messages. We looked over all the takeaways and made some changes to more precisely communicate the exact thing of the preceeding paragraph. Concretely, we replaced the takeaway message you mentioned with the following instead:
>
> > With a mild decay assumption on the learned rank-1 components, we can bound the performance change between ranks which ensures that the compute–performance trade-off remains smooth and predictable even for untrained ranks.
>
> **Actions taken.** Made the takeaway message less repetitive (**L314-316**)
>
> ---
>
> ## (I) How the anchor and variant rank are chosen?
>
> Let us clarify how the ranks are chosen. We distinguish between two types of ranks: the *anchor rank* (which is the highest possible rank), and a *variant rank* which is a smaller rank, typically between 1 and the maximum (anchor) rank. We made this choice because: (i) including the anchor rank helped to stabilize training and learn the best performance on the highest rank; and (ii) the variant rank helps to learn
>
> **How are the variant ranks chosen?** For each batch, we always run two forward passes: one with the anchor rank and one with a sampled rank which is chosen from a pre-defined list of ranks. As a simple example, if our maximum rank is 64, we might choose our variant ranks as `variant_ranks = [1, 2, 4, 8, 16, 32]`, and a single batch would have two losses computed: (i) the loss at rank 64; and (ii) the loss at the sampled variant, which would be randomly sampled from that list. These two losses are then weighted (described in **L212-215**). We also show that our model generalizes to variant ranks we have not chosen at training time (e.g. for ranks ``[2, 3, 5, ...]``) (**(Table 1)**).
>
> Therefore, there is randomness in how the variant rank is chosen for each batch of data.
>
> We have also added a full algorithm in the main paper and added an explanation as follows:
>
> >  Algorithm 4 summarizes the full training procedure. Each iteration evaluates the model at the maximal anchor rank and at a sampled variant rank, combines their losses through rank-specific uncertainty weights, and updates both the shared parameters and the log-variances. This mechanism jointly optimizes all submodels and stabilizes learning across heterogeneous ranks.
>
> We have made this clearer in the writeup.
>
> **Actions taken**.  (1) Added pseudocode in the writeup to explain the anchor and variant usage **(L216-226)**.; (2) Explained the algorithm in words (**L247-250**); (3) Added full pseudocodes of forward passes of layers, networks, and training (**Appendix E**)

---

> > ### Author Response · Authors · 2025-11-20
> > **Response (4/4)**
> >
> > ## (J) Log-variances serve as an emergent proxy for expresiveness of each rank
> >
> > We appreicate the question on why log-variances serve as an emergent proxy for parameters.
> >
> > The fact that the learned log-variances act as a proxy for expresiveness is in fact one of the reasons why we're so excited by this method/framework due to this resulting interpretability.
> >
> > **Are the log-variances free parameters?**
> > The log-variances are not *free* parameters, yet they are *trainable* parameters. The reason they are not *free* is because in the objective $e^{-s_k}\mathcal{L}_{\text{CE}}(k) + s_k$, each log-variance $s_k$ is coupled to the rank-$k$ loss. This means their values are directly dependent on the loss within each model with a given rank. But we still *learn* this value.
> >
> > **So, why does log-variance serve as an emergent proxy?**
> > **Short answer:** The trick is to understand that this parameter basically tracks the "residual loss" for each rank. This means that higher residual loss (which means higher error for a given rank) will have a higher learned uncertainty parameter. Therefore, this parameter serves as an emergent proxy for expresiveness, as higher residual loss implies a less expressive model, and this parameter is tracking this residual loss.
> >
> > **Full explanation why log-variances are proxies for expressiveness (also in Appendix D.6 - D7.)** (Unfortunately, openreview seems to have rendering issues with formulae and we had to re-write the below in words. Appendix D7 gives a more rigorous treatment of the same answer).
> >
> > First, each rank-model contributes differently to the training objective. For a model of rank *k*, its contribution has two components: one term that depends on the cross-entropy at that rank, scaled by the negative exponential of its log-variance, and another term that is simply the log-variance itself. In other words, the objective combines “how well the rank fits the data” with a learned uncertainty term.
> >
> > Second, consider what happens once training settles into a stationary point (a point where gradients vanish). At that point, the condition on the log-variance simplifies dramatically: the exponential of the negative log-variance must match the inverse of the residual cross-entropy. Rearranging this condition shows that the log-variance ends up equal to the logarithm of the remaining cross-entropy at that rank (please consider seeing Appendix D7 if this doesn't make sense: openreview seems to not render the equations).
> >
> > This means that, aside from optimization noise and parameter interactions, the **learned log-variance mirrors the scale of the leftover error at that rank**.
> >
> > **Now compare a “more expressive’’ to a “less expressive’’ model.** A more expressive model can push its cross-entropy lower, which forces its corresponding log-variance downward. A less expressive model cannot reduce its residual error as much, so its log-variance remains higher. Across training, this mechanism enforces a simple ordering: ranks that capture the data well develop lower log-variances, and ranks that struggle develop higher ones.
> >
> > **On the uncertainty objective**
> >
> > The same algebraic structure appears in uncertainty-based multi-task weighting, where the optimal variance increases with the irreducible error of a task; by analogy, each rank behaves like a “task” whose residual error determines its effective variance, which makes the learned variances function as uncertainty surrogates.
> >
> > **Actions taken.** We've added a full expalanation (**Appendix D.6.**) and (b) added a short explanation in the paper **(L228-235).**
> >
> > ---
> >
> > ## (K) Source code
> >
> > Yes, we: we have all the source code ready to be released upon the paper being accepted. This includes:
> > - The drop-in layer replacements
> > - An easy-to-use function to replace any language model's layers with rank-adaptive layers we introduce in the paper
> > - Code to reproduce all experiments
> > - Notebooks with figures
> >
> >
> > ---
> >
> > ## Thank you
> >
> > You have helped us improve our paper - thank you.
> >
> > If you have any other concerns or questions, we would be happy to address them. We are very excited by the developments this architecture can bring to making models natively performane-compute adaptive. We will of course open source all the code upon acceptance.
> >
> > If you feel the paper has improved as a result and that the deep learning community would benefit from this method, we kindly invite you to consider re-assessing the current rating.

---

> ### Comment · Reviewer_TDZV · 2025-11-26
>
> Thank you for your extensive responses. They have clarified my questions. I find the updated manuscript much improved in clarity so I am raising my score from 4 to 6.

---

### Official Review · Reviewer_bkcs · 2025-10-31

**Soundness:** 3
**Presentation:** 3
**Contribution:** 3
**Rating:** 6
**Confidence:** 3

**Summary:**

This paper proposes Nested Subspace Networks (NSNs), an architectural  framework for neural networks that enables continuous and dynamic  adaptation to a range of computational budgets through a nested subspace reparameterization of linear layers. NSNs introduce a means to  dynamically select model "rank" at inference, with theoretical backing  for smooth interpolation across ranks and an uncertainty-aware  multi-rank training objective to promote optimality across the entire  rank spectrum. Empirical evaluations include MLP experiments on CIFAR-10, ablation studies, and applications to several large pre-trained language models (LLMs), demonstrating that NSNs achieve competitive accuracy with significantly reduced computational cost.

**Strengths:**

- The paper proposes a conceptually novel framework, Nested Subspace Networks, that unifies multiple model capacities within a single parameterization.
- The approach is validated through controlled experiments, ablation analyses, and applications to pre-trained LLMs, supporting the claims of dynamic adaptability and compute–performance efficiency.
- A key practical merit is the post-hoc applicability of NSNs: the framework can be retrofitted onto existing pre-trained models without retraining from scratch, addressing a critical deployment challenge for large-scale foundation models.

**Weaknesses:**

- Equation 2 needs clarification. Does the variant rank $r$ change in different epoch or remain unchanged during training. Is $\mathcal{L}_{CE}(k)$ the cross entropy loss calculated when the linear weight using its first $k$ rows and columns?
- Implementation details missing. The paper does not explain how Logits Regularization, Residual Orthogonality, and Hidden Regularization (as shown in Table 1) are implemented, nor whether these are used concurrently with the main “Two CEs” objective or as independent ablations.
- Ambiguity in Section 4.1. The experiment uses CIFAR-10 but states that “inputs are ImageNet last-layer embeddings.”
Please clarify this setup: are ImageNet features used merely as a fixed embedding extractor, and are all MLP layers subsequently replaced with NSN layers?

**Questions:**

1. Why choosing exponential in Equation 3 as the coefficient.

2. Is there any empirical evidence about assumption 1.

3. In Figure 7, it is shown that the model's score on the evaluation  benchmarks decreases continuously and monotonically as the rank is  reduced. How can we prove that this decline represents the desired  hierarchical degradation in reasoning ability, rather than simply a  score reduction caused by a decrease in the model's overall expressive  power?

For instance, is the score drop a result of the model becoming  inconsistently capable of solving tasks of the same difficulty level  (i.e., succeeding on some while failing on others)? Or is it due to the  model becoming uniformly incapable of solving a whole category of  problems at a certain difficulty level? Only the latter would  demonstrate an effective adjustment of reasoning and computational  effort according to task difficulty.

4. A recent paper [1] on implicit regularization in matrix factorization (which is not cited in Related Works) also analyzes the hierarchical training dynamics that emerge when progressively increasing the matrix rank—an idea conceptually related to the nested structure proposed in this paper (see, for example, their Proposition 1). One of their key findings is that even without explicit rank constraints, the model naturally evolves from low-rank to high-rank representations during optimization. It would strengthen the theoretical positioning of this work if the authors could discuss or compare the necessity of explicitly enforcing rank constraints in NSNs.

[1] Connectivity Shapes Implicit Regularization in Matrix Factorization Models for Matrix Completion, NeurIPS 2024.

---

> ### Author Response · Authors · 2025-11-20
> **Response (1/3)**
>
> Dear reviewer bkcs,
>
> Thank you for taking the time to carefully review our work (it's clear you took the time to read this thoroughly - sincerely thank you). We're glad that you agree that NSNs are conceptually novel, validated via ablations, experiments, and that they, importantly, enable post-hoc applicability to larger models.
>
> **Brief summary**. You rightly asked for more precise exposition and stronger theoretical positioning. We have therefore **(i)** clarified Equation (2) and the role of anchor/variant ranks via explicit algorithms, **(ii)** fully specified how the different regularizers in Table 1 are instantiated and used, **(iii)** added empirical tests of Assumption 1 and violation statistics to substantiate the interpolation bound, and **(iv)** extended the related‑work discussion to connect NSNs to implicit regularization in matrix factorization and to explain why explicit rank constraints are still needed in deep networks. With these additions, we hope we have fully addressed your concerns; and we hope you consider re-assessing the paper to support it being featured at ICLR.
>
> We have also added a summary of the **15 experiments** we've conducted in the paper, as well as the **11 changes to the paper** after the rebuttal in the global response which we encourage you to take a look.
>
> We now address your points specifically in sections **(A)-(G)**
>
> ## (A) Clarifying Equation 2
>
> Thank you for the question and for highlighting the need to clarify Eq. 2. Your understanding is correct on both.
>
> **On whether the rank varies during training.**
>
> Yes — the rank is sampled and varied throughout training. At each training step we draw a rank $r$ from the predefined rank set from the predefined rank set (according to the uncertainty-aware sampling distribution we introduced). The linear layer is then instantiated at that sampled rank for the forward and backward pass. In other words, the model is trained across the entire family of ranks rather than with a single fixed rank.
>
> **On whether the loss is calculated using the first $k$ rows or columns**
>
> Yes, your understanding is correct: For a sampled rank $r$, we form the corresponding subspace parameters by taking their first $r$ columns (and rows for square matrices) of weight matrix. The cross-entropy loss (for a single rank) is computed on the output of this rank-$r$ instantiation. We total cross entropy at each iteration is then the highest rank and a sampled variant rank.
>
> To make this clearer, we've made four changes in the paper.
>
> **Actions taken**. (1) We have added a more intuitive explanation under Figure 1 (**L69-74**); (2) We've added the training procedure and loss computation as a separate Algorithm in the paper (**L216-226**); (3) We added a paragraph to explain this logic in the main paper more clearly (**L247-250)**. (4) We added the full pseudocode for the forward pass and loss computation (**Appendix E**).
>
> ---
>
> ## (B) Adding implementation details
>
> We appreciate you pointing this out -- we somehow missed this in the final version. We've now fixed this.
>
> Your understanding is correct: all three variants—Logits Regularization, Residual Orthogonality, and Hidden Regularization—are implemented as independent ablations, each adding exactly one regularizer on top of the same anchor/variant "Two CEs" training setup. The "Two CEs" is our method, where the first CE loss is the anchor rank and the second is the variant rank.
>
> **Actions taken** (1) We have added a summary in the main paper (**L358-361**); (2) We have included all the relevant details to fully reproduce the logic (**Appendix B2**).
>
> ---
>
> ## \(C) Clarifying CIFAR-10 usage (Section 4.1)
>
> Your interpretation is exactly right. We use ImageNet features as a fixed embedding extractor, and all MLP layers are replaced with NSN layers. We've made this less ambiguous now in the text.
>
> **Actions taken**. (1) Removed ambiguity in explanation in the main text (**L335-342**); (2) Added a step-by-step walkhrough to supplement this (**Appendix B.1.**)

---

> ### Author Response · Authors · 2025-11-20
> **Response (2/3)**
>
> ## (D) Why choose exponential in Equation 3 as the coefficient?
>
> We appreciate the clarification. We think this explanation should be present in the main paper which we have now added. The following is now added right after including the equation (**L228-235**)
>
> >  Why are the exponentials useful in this equation? It's useful to reason about this from three different perspectives. First, the exponentials are useful because the reparameterization $w_k = e^{-s_k}$ ensures strictly positive weights and produces a strictly convex objective in $s_k$. This is useful, since it provides stable gradient updates. Second, this formulation yields a closed-form optimum $w_k^\star = 1/L_k$. This is useful because we (i) become scale invariant and (ii) have gradient balancing across ranks of different difficulty. Third, this is directly tied to building surrogates that are based on Gaussian regression likelihood for classification settings which are easy to optimize. More details in Appendix D6.
>
> **Actions taken**. (a) Added explanation in the main paper; (b) Expanded the logic and derivation in the Appendix (**Appendix D6**).
>
> ---
>
> ## (E) Empirical Evidence for Assumption 1
>
> In response to your question on whether there is empirical evidence for Assumption 1, we ran a set of experiments. We have found that this assumption does hold in practice.
>
> **Setup**. We took a GPT-NeoX model trained with our recipe and looked at how the energy (measured by the Frobenius norm) distributes across different ranks. This gives a direct picture of whether the assumed decay appears in practice.
>
> **Results**.  We have found that the assumption holds in practice.We in fact plot multiple sub-figures and show that the decay is particularly pronounced in the first indices and the rate of decay gradually decreases. This, of course, makes sense, as the first base indices must encode most of the information. (**Fig 11 (a) and Fig 11 (b)**)
>
> **Further results**. To make the picture more precise, we also added a layer-by-layer table reporting how often the assumption is locally violated, if at all (i.e., are there any violations to this assumption?). This table is reported in **Table 4**. We quantified that this happens in 0.04\% of cases (not 4\%) of all basis vectors. This means that 99.96\% of the time, this assumption holds. Because these violations to the assumption are sparse and small, and given how consistently this assumption holds, we interpret them as noise in the optimization process. them as minor fluctuations introduced by the optimization dynamics
>
> **Actions taken**. (1) Updated main paper to include the empirical support for this assumption (**L295-297**); (2) We show the empirical support for this, including the first indices with short-horizon energy decay **(Fig 11a)** and the full-range energy decay **(Fig 11b)** (3) We've included a separate empirical analysis of when this assumption can be violated (**Table 11**); (4) We've added a separate subsection to discuss these findings (**Sec C.4**).
>
> ---
>
> ## (F) Fig 7: Does the decline represent the desired hierarchical degradation?
>
> We do not make any claim in the paper that our low-rank model represents different "reasoning" levels. We understand that there *is* a distinction between "hierarchical degradation in reasoning ability" versus "a score reduction caused by a decrease in the model's overall expressive power", but we make no claim about whether or how this corresponds to any of these two mechanisms or that lower-rank subspaces correspond to distinct "reasoning" levels.
>
>
> Instead, we focus only on the empirical and theoretically expected property that reducing rank yields a smooth, monotonic decrease in performance, and we show that the nested parameterization consistently enforces this relationship across architectures and benchmarks; this alone is the claim we support, and it is fully demonstrated by the observed behavior.
>
> Having said that, exploring whether rank could be linked to meaningful tiers of problem difficulty or to structured patterns of capability loss is an interesting extension, but requires experiments—such as task-stratified evaluations or compositional diagnostic suites—that go beyond the scope of the present work.
>
> We also think this is a non-trivial problem, since it requires to operationalize "reasoning", disentangle effects of different sub-layers and do mechanistic interpretability-type work.
>
> **Actions taken.** We've updated the first section of the introduction (**L32-39**) to better position what problem we attempt to solve.

---

> ### Author Response · Authors · 2025-11-20
> **Response (3/3)**
>
> ## (G) On the implicit regularization in matrix factorization
>
> Thank you for providing this reference - we hadn't noticed this work before. We answer the two sub-questions you had.
>
> **(a) Is this related work?**
> We have now added this paper to our related work section (**L838-840**).
>
> **A brief summary** The papers are related in that they operate in the low-rank regime but are different because, in the work you've cited, the authors work on a model which is itself a low-rank parametrization $W=UV^T$. So, any gradient-based optimization is confined to that low-dimensional manifold; the training dynamics can only move within this factorized space, which makes a progression from effectively low to higher rank unavoidable as the factors grow during optimization. In contrast, NSNs operate in full deep networks whose linear layers are unconstrained full-rank matrices unless we explicitly re-parameterize them.
>
> **(b) Do full-rank layers produce hierarchies without constraints?** You've asked whether this can actually happen without our regularization. We have now run three more experiments to verify that this cannot happen and show why NSN produce a unique arrangement of matrix information.
> - First, we have  tested this empirically: we have computed the same energy decay from point (F) on a naturally trained MLP model under cross-entropy loss (**Sec C.5**). **We find that these hierarchies do not form under normal fine-tuning**.
> - Second, we run more experiments to quantify how often such of "energy decay" occur in normally trained vs Nested Subspace Networks (**Fig 13 and Fig 14**). We find that violations to Assumption 1 occur almost 50\% of the time in normal models and less than 1\% of the time in NSNs. This supports our claim that under normal fine-tuning, there is no implicit ordering of basis vectors.
> - Third, we also show this in the main paper. For instance, Figure 2 shows that simple truncation does not induce a sufficiently good training scheme.
> - Fourth, we have run additional experiments to show why this happens (**Fig 10**). We show that this happens because NSNs arrive at different solutions in the optimization landscape relative to standard fine-tuning. We show this by quantifying the cosine similarity between NSN fine-tuned weights and dense-model fine-tuned weights and find that the highest cosine similarity is .85. This shows that the methods never converge to the same solution given the same initialization.
>
> Therefore, we have ample evidence to show that standard fine-tuning cannot arrive at the same solution as the optimization procedure proposed in the paper.
>
> We have updated the paper to reflect these comments.
>
> **Actions taken**. (1) Added new results: empirical comparison to show energy decay does not exist in normal models (**C5**); (2) Added new results: violation comparison between NSN and regular dense models (**Fig12-14**); (3) Added an explanation in the main section (**L296-297**); (4) Updated related work to include your reference **L838-839**.
>
> ---
>
> ## Thank you
>
> You have helped us improve our paper - thank you.
>
> If you have any other concerns or questions, we would be happy to address them. We are very excited by the developments this architecture can bring to making models natively performane-compute adaptive. We will of course open source all the code upon acceptance.
>
> If you feel the paper has improved as a result and that the deep learning community would benefit from this method, we kindly invite you to consider re-assessing the current rating.

---

### Official Review · Reviewer_Ji5G · 2025-11-05

**Soundness:** 2
**Presentation:** 3
**Contribution:** 1
**Rating:** 2
**Confidence:** 4

**Summary:**

This paper presents a model architecture called Nested Subspace Networks (NSNs), which can be applied on top of modern neural network architectures both during pretraining and after pretraining. NSNs consist of a series of low rank subnetworks that approximate the original neural network, thereby reducing computational FLOPs during training, adaptation, and inference. A novel optimization procedure based on uncertainty estimation is also proposed to jointly optimize all subnetworks within the NSN architecture. Both theoretical analysis and experimental results are provided to demonstrate the effectiveness of the proposed NSNs and the optimization method.

**Strengths:**

- The paper is overall well written and well motivated.
- Improving the computational efficiency of large deep models and large language models is of practical importance.
- The proposed NSN model architecture is clear and easy to follow.

**Weaknesses:**

- The NSN architecture does not appear to be highly novel. It essentially applies the slimmable neural network [1] on top of a low rank model architecture (across the rank dimensions).
- The proposed "training with multi rank uncertainty" procedure is not clearly explained. I suggest adding an algorithm box to illustrate the detailed training steps.
- Only FLOPs are reported; wall clock time would be a more informative metric to demonstrate both training and inference speedups.
- It is not clear, during model training, what the overall cost of training an NSN is compared to the standard training of a dense model with a similar number of parameters.

[1] https://arxiv.org/abs/1812.08928

**Questions:**

- The methods proposed in [2] seem to be very promising on top of slimmable networks. I wonder if they are also applicable to NSNs.


[2] https://openaccess.thecvf.com/content_ICCV_2019/papers/Yu_Universally_Slimmable_Networks_and_Improved_Training_Techniques_ICCV_2019_paper.pdf

**Details Of Ethics Concerns:**

The reviewer does not seem to find any ethics concerns of this paper.

---

> ### Author Response · Authors · 2025-11-20
> **Response (1/4)**
>
> Dear Reviewer Ji5G,
>
> Thank you for your throughtful comments to improve the paper. We provide a brief summary below; and concrete answers (A) - (F) and highlight updates to the paper.
>
> **Brief summary**. Your main reservations were that NSNs might be little more than "slimmable networks over rank" and that the multi‑rank uncertainty training and computational benefits were not specified precisely enough to justify a new architecture. In the revision we therefore (i) introduce a dedicated comparison to slimmable / universally slimmable networks and LoRA across multiple criteria to show that they are entirely unrelated classes of models. We've added other things to address your concerns, such as new algorithmic pseudocode and a clearer derivation of the uncertainty‑weighted objective; we address all your points below. Our goal is that, with these clarifications and the strengthened theory/diagnostics, the contribution is clearly seen as a distinct and practically relevant framework.
>
> We have also added a summary of the **15 experiments** we've conducted in the paper, as well as the **11 changes to the paper** after the rebuttal in the global response which we encourage you to take a look.

---

> ### Author Response · Authors · 2025-11-20
> **Response (2/4)**
>
> ## (A) Differences from slimmable neural networks
>
> We appreciate your concern about the differences from slimmable neural networks. We believe there may be a misunderstanding about this: while slimmable neural networks solve a similar problem, the *properties* of the two paradigms are very different. In fact, we highlight these differences in our paper, including the abstract (**L15-17**), introduction (**L 51-52**), related work (**L472-475 and L497**).
>
> To clarify: **we do not apply a slimmable neural network on top of a low rank model architecture**. Slimmable Nets suffer from major problems that NSNs are designed to solve. In fact, the limitations of slimmable neural networks is what motivated us to design our method in the first place.
>
> Just to name a few key limitations that slimmable neural networks have:
> - (1) Slimmable Nets have extremely large training cost which scale with the number of chosen "widths"; our method does not because we only sample two ranks at any given point (**L247-250**); *Practical importance*: If we create 100 different subnetworks, this will incresae the *training costs* 100x with Slimmable Nets; and it will not increase the training costs with Nested Subspace Networks.
> - (2) Slimmable Networks can only be applied to pre-chosen widths; our method supports arbitrary ranks (**Sec. 3**); *Practical importance*: Choosing 5 widths with slimmable nets gives only 5 "subnetworks". In contrast, NSNs provide a continuum of networks, even for ranks not even trained before. We're not limited to a few pre-selected networks.
> - (3) We cannot apply or use Slimmable Networks for pre-trained large models without redesigning the models or training from scratch; our work allows us to do that (**Sec 4.4**); *Practical importance*: This is in fact why we designed this method to begin with: we were working with large pre-trained foundation models and wanted to make it adaptable to different computational budgets without expensive re-training of a big foundation model.
> - (4) Slimmable Networks do not guarantee smooth trade-offs between widths and performance, our method does by construction of the theory (**Sec 3**).
>
> **Apart from these major differences, our work answers many important questions that Slimmable Networks do not address**, such as: (a) how do we ensure you we perform inference at any budget? (**Sec 2.2**); (b) We never train for *all* computational budgets; how can we obtain guarantees that the performance will be good for untrained budgets? (**Sec 3**); \(c) What regularization strategies are important/suficient for us to get good compute-performance frontiers? (**Sec 4.2**; (d) How to adapt such a model to pre-trained foundation models? (**Sec 4.3**); (e) Can we validate this behavior on actual LLMs and demonstrate empirical performance frontiers? (**Sec 4.1-4.4**).
>
> We appreciate this might not have been clear. We have made these changes:
> - Added an explanation on adaptation to existing models and architectural differences (**L122-126**)
>
> > Unlike slimmable networks \citep{yu2018slimmable}, which vary channel width and therefore change intermediate tensor shapes, NSNs only vary the rank of a shared low-rank factorization, so all input–output dimensions of each layer remain fixed and the architecture can be inserted into pre-trained transformers and LLMs without modifying their interfaces or normalization layers.
>
> - Added more architectural differences (**L787 - 793**)
> > In contrast to slimmable networks, whose behavior between trained widths is controlled only empirically through regularizers such as the sandwich rule, the nested subspace structure of NSNs lets us bound the change in expected loss between any two ranks, yielding a theoretically controlled interpolation between compute budgets.
>
> - Added a separate appendix section discussing this (**Lines 787-796**). In fact, you motivated us to significantly expand our related work section which we have now expanded (**Appendix A**).
>
>
> We have now fixed this.
>
> **Actions taken**. (a) Added new explanation in the main text contrasting slimmable and NSNs; (b) adding in-depth explanation appendix; \(c) creating new related work appendix

---

> ### Author Response · Authors · 2025-11-20
> **Response (3/4)**
>
> ## (B) Can we apply the methods from Universal slimmmable neural networks on top of NSNs?
>
> You mentioned that the methods proposed in [2] (universal slimmable neural networks) seem to be promising on top of NSNs. We appreciate the suggestion. We in fact compare our method to the work you cited in the paper already against our work **(Table 3)**, as this is a foundational piece in the literature.
>
> **Universal slimmable neural nets cannot complement NSNs, since we operate with different, orthogonal mechanisms**. Specifically, our work is designed to solve key issues that appear in universal slimmable nets:
>
> - (1) Slimmable universal Nets are trained from scratch, so they also cannot be adapted to large pre-trained foundation models which significantly limits their availability (**L416-429 and Table 3**). **What this means in practice?** Applying their model to, for example, a 70B LLM is not realistic, as it changes the entire architecture (widths, batch-norm behavior). *Our work is the first work to introduce a dynamic technique that can be retrofitted on any pre-trained foundation model.* This is extremely important, since many top models today are large pre-trained expensive foundation models.
> - (2)  Slimmable Universal Networks rely on tricks such as inplace distillation or sandwhich rules. Our work shows that these are not required and give an expalantion as to why **(Sec 4.2 and Sec 2.2)**. This is a standalone contribution: we show that there is no need for custom, specific ad-hoc tricks.
> - (3)  Our model *guarantees* smooth performance because of the nested function classes (**Sec 3**). Slimmable Universal Networks do not possess such guarantees. This is a key architectural difference which enables to obtain different properties. This also gives us higher-levels of interpretability as we can directly reason about the hierarchy of sub-models as nested function classes, borrowing from a very rich literature.
> - (4) Lastly, Universal Slimmable nets are not architecturally agnostic. Specifically, they alter channel dimensions, which changes output shapes and breaks compatibility with accelerators that depend on fixed, ahead-of-time–compiled tensor graphs; this makes their theoretical flexibility impractical because any width or depth change requires costly or infeasible recompilation. **What does this mean in practice?** Using their model in practical settings requires recompiling or deploying whole binary codes which makes it almost impossible to use on user devices. Our approach avoids this entirely: NSNs keep all tensor shapes fixed and adapt compute only by selecting different numbers of basis elements.
>
> This is why we are so excited by this work -- it's directly addressing issues existing in prior work.
>
> To make this clear, we've implemented a few changes to ensure there can be no ambiguity about this.
>
> **Actions taken**. To address this, we have already: (a) Extended the related work discussion to highlight the above differences (**Appendix A**); (b) Added more commentary in **Sec 2.1**.
>
> You raising this question has been very productive and useful for structuring our contributions even more clearly -- we appreciate this.
>
> ---
>
> ## \(C) Clarifying the multi-rank uncetainy procedure
>
> We appreciate you pointing out that our current explanation of multi-rank uncertainty is not clear. We've resolved this.
>
> **Actions taken** (a) Added a full algorithm pseudocode in the appendix (**L216-226**); (b) Added a summary of that pseudocode in the main text (**L236-243 and L247-250**); \(c) We will of course release the code publicy upon acceptance.
>
> ---
> ## (D) Reporting wall clock time
>
> We appreciate your suggestion to report wall-clock time. There are two primary reasons why reported FLOPs and did not add wall-clock time:
> - First, wall-block time is the standard in the adaptive compute literature: it is hardware-agnostic, does not depend on runtime configuration, system factors, caching, threading, background load, or other common system factors. It's a very useful metric.
> - Second, this is the primary measurement across related work, including the Slimmable Nets related work you cited (we double checked -- none of those works report wall-clock time).
>
> However, we understand that obtaining wall-clock time can be useful. We have re-ran the FLOPs experiments and found that wall-clock time on our H100 GPU was directly correlated with the FLOPs (with +- 5\% deviation), so the "time savings" percentage-wise gave the same signal. We've updated the paper to reflect this. (As a side note, we can convert time FLOPs to wall-clock time with a formula: $\frac{\text{total FLOPs}}{\text{Effective hardware FLOPs}}$, so the wall-clock time might be differ for different users)
>
> **Actions taken**. Explained motivation for measurement in FLOPs (**L1566-L1579**).

---

> ### Author Response · Authors · 2025-11-20
> **Response (4/4)**
>
> ## (E) Cost of training an NSN compared to standard nets
>
>  Training an NSN requires two forward–backward passes per step—one at the anchor rank and one at a smaller variant rank—which makes each step up to about twice as expensive as training a single dense model. Despite this overhead, an NSN replaces the need to train many separate models when multiple computational budgets are required. This means the total cost becomes far lower than training several dense networks, because one NSN training run supports all desired operating points. This was implied by the fact that there are two ranks (and therefore two forward passes) during the training time, but we now made it explicitly clear in the paper.
>
>  We've added a paragraph in the paper to address this:
>
> > Compared to a standard dense network with weight matrix $W \in \mathbb{R}^{d_{\text{out}} \times d_{\text{in}}}$, whose per-step training cost is proportional to $d_{\text{in}} d_{\text{out}}$ FLOPs, training a Nested Subspace Network (NSN) layer with maximum training rank $\tilde R$ and a sampled variant rank $r < \tilde R$ requires per-step FLOPs proportional to $(\tilde R + r)(d_{\text{in}} + d_{\text{out}})$, because each optimization step performs a forward–backward pass at the anchor rank $\tilde R$ and another at the variant rank $r$.
>
> > Using the break-even rank $R_{\text{be}} = \frac{d_{\text{in}} d_{\text{out}}}{d_{\text{in}} + d_{\text{out}}}$, for which a single low-rank pass matches the dense cost, the pessimistic case $r \approx \tilde R \approx R_{\text{be}}$ gives a total of about $2 d_{\text{in}} d_{\text{out}}$ FLOPs per step, i.e., at most roughly twice the cost of training one dense model with the same input and output dimensions.
>
> > However, this single NSN training run yields a whole hierarchy of usable ranks at test time, so if $K$ different computational budgets are needed, the NSN still replaces $K$ separate dense training runs (total cost $\approx K d_{\text{in}} d_{\text{out}}$) with one run whose cost is only a constant factor above that of a single dense model, effectively amortizing the training cost over many operating points.
>
>
> **Actions taken**. (a) Added Algorithm 1 that showcases inference costs (**L216-226**); (b) Added a new appendix (**L1566-L1579**) that directly evaluates training costs of NSNs and standard nets.
>
> ---
>
> ## (F) Summary of key contribution
>
> We hope we've addressed your concern on slimmable neural networks. To reiterate, our method is in no capacity using slimmable neural network architecture or ideas. In fact, it was designed to address key limitations stemming from the method. We hope this is now clear, and we have implemented all the changes discussed in this response.
>
> We're very excited about this work because this enables deep learning practitioners to deploy compute-performance adaptive models in a variety of settings.
>
> Thank you for your help in improving the paper.

---

### Author Response · Authors · 2025-11-20
**Global response (1/2): Summary of experiments and findings**

We thank the reviewers for their insightful and constructive feedback.

Reviewers highlighted that the paper is “overall well written and well motivated” (**R-Ji5G**) and proposes a “conceptually novel framework that unifies multiple model capacities within a single parameterization” (**R-bkcs**). They emphasized that “improving the computational efficiency of large deep models and large language models is of practical importance” (**R-Ji5G**), and viewed the approach as “highly flexible and practical as it can be broadly applied to any pretrained models” (**R-TDZV**).

The NSN architecture itself was described as “clear and easy to follow” (**R-Ji5G**) and “interpretable in the sense that it is straight-forward to inspect how much each rank learns to represent both empirically and theoretically,” with the theoretical discussion providing “good intuition of the approximation trade-offs involved” (**R-TDZV**). Empirically, reviewers noted that the approach is “validated through controlled experiments, ablation analyses, and applications to pre-trained LLMs, supporting the claims of dynamic adaptability and compute–performance efficiency” (**R-bkcs**). They further highlighted as “a key practical merit” that NSNs are “post-hoc applicable” and “can be retrofitted onto existing pre-trained models without retraining from scratch, addressing a critical deployment challenge for large-scale foundation models” (**R-bkcs**).

---
# Summary of experiments and findings

We'd like to highight the **15 experiments** we've conducted as a summary. We hope these will clearly showcase the value of Nested Subspace Networks.


| Experiment | Figure/Table | Purpose | Finding |
|-----------|--------------|---------|---------|
| Native rank training vs rank truncation on CIFAR-10 MLPs | Figure 2 | Compare truncation with native low-rank training. | Native low-rank training greatly outperforms naive truncation. |
| Log-variance dynamics across ranks | Figure 3 | Examine uncertainty weights across ranks. | Uncertainty weights differentiate ranks according to difficulty. |
| NSN vs individually trained CIFAR-10 MLPs | Figure 4 | Test NSN against specialized MLPs across ranks. | A single NSN matches specialized models across ranks. |
| Ablation of multi-rank training objectives | Table 1 | Compare alternative multi-rank training objectives. | Two CEs objective improves low and intermediate ranks. |
| Training dynamics at interpolated ranks | Figure 5 | Assess stability at interpolated (untrained) ranks. | Two CEs stabilizes accuracy at interpolated ranks. |
| Post-hoc adaptation of pre-trained LLMs | Figure 7 | Evaluate NSNs’ accuracy–FLOPs trade-offs on LLMs. | NSNs yield smooth, controllable accuracy–FLOPs trade-offs. |
| Inter-layer similarity vs rank in NSN-adapted LLM | Figure 8 | Analyze inter-layer similarity across ranks. | Low ranks share representations; higher ranks specialize by layer. |
| Subspace containment across ranks | Figure 9 | Test nested subspace property empirically. | Lower-rank subspaces are nearly contained in higher-rank subspaces. |
| NSN vs standard dense fine-tuned model | Figure 10 | Compare NSN weights to dense fine-tuned weights. | NSN weights differ systematically from dense fine-tuned weights. |
| Energy decay in NSN rank-1 components | Figure 11 | Check energy decay across NSN components. | NSN components show clear energy decay across indices. |
| Energy profiles in dense GPT-NeoX layers | Figure 12 | Inspect energy profiles in dense layers. | Dense models lack ordered energy decay structure. |
| Aggregate energy-decay violation rates | Figure 13 | Quantify energy-order violations in NSN vs dense. | NSNs almost never violate energy-decay ordering; dense models do. |
| Violation rates by transformer depth | Figure 14 | Study violations as a function of depth. | Dense models show high violation rates across depths; NSNs do not. |
| Layer-wise violation statistics | Table 4 | Provide detailed per-layer violation statistics. | Most NSN layers have very few energy-order violations. |
| FLOPs–accuracy trade-off via surgical replacement (Pythia-2.8B) | Figure 15 | Quantify Pythia-2.8B FLOPs–accuracy trade-off. | Halving FLOPs costs roughly five accuracy points on Pythia-2.8B. |

---

### Author Response · Authors · 2025-11-20
**Global response (2/2): Changes after the rebuttal**

# Changes after the rebuttal
In the uploaded PDF, we have highlighted the changes in the main text with a bright distinct color. We did not highlight changes in the appendix. **We implemented 11 changes that address all the reviewers' concerns**.

**Major change summary.** We now make explicit that NSNs are not *slimmable low-rank layers* but a parameter-space reparametrization that enforces a nested sequence of subspaces in every linear layer, jointly trained with a principled multi‑rank objective so that **(i)** the function class at rank $r$ is a strict subset of that at rank $r +1$; **(ii)** performance between trained ranks is theoretically controlled via an interpolation bound; and **(iii)** a single model provides a continuous, monotone compute–performance frontier that can be surgically retrofitted onto pre‑trained LLMs. We complement this with: **(iv)** an explicit training algorithm, **(v)** a complexity analysis of NSN training vs dense baselines, and **(vi)** additional empirical diagnostics (energy decay and subspace containment in LLMs), so that the paper now provides a complete, theoretically grounded and practically usable recipe for turning existing large models into compute‑adjustable ones.

We summarize the major changes here:

| Change | Purpose | Reviewer(s) addressed | Summary |
| --- | --- | --- | --- |
| Clearer motivation and revised intro (Sec. 1) | Remove ambiguity and give concrete deployment scenarios | TDZV | Added three specific use-cases and replaced unclear phrasing. |
| Clarified NSN architecture, definitions, and visualization (Sec. 2.1, Fig. 1, App. D.1) | Make the mechanism and nesting explicit | TDZV, Ji5G, bkcs | Added explicit A/B explanation, corrected figure, and added toy matrix example. |
| Added full training algorithms (Alg. 1; Algs. 2–4 in App. E) | Specify training steps and rank sampling | Ji5G, bkcs, TDZV | Introduced pseudocode detailing anchor/variant ranks and updates. |
| Expanded explanation of uncertainty-weighted objective (Sec. 2.2–2.3, App. D.2, D.6) | Explain weighting, exponentials, and balancing | bkcs, TDZV | Added derivation and interpretation of exp(−s) and gradient balancing. |
| Added empirical validation of Assumption 1 (Sec. 3; App. C.4–C.5, Table 4) | Support smooth interpolation theory | bkcs | Added energy-decay plots and violation-rate measurements across layers. |
| Clarified CIFAR-10/MLP setup (Sec. 4.1, App. B.3) | Resolve confusion about embeddings and baseline choices | bkcs, TDZV | Specified ImageNet-feature pipeline and rationale for excluding LoRA. |
| Added implementation details for regularizer ablations (Sec. 4.2, App. B.2) | Clarify how each regularizer is applied | bkcs | Documented each regularizer and confirmed they are never combined. |
| Extended Related Work + new Appendix A | Better distinguish NSNs from prior methods | Ji5G, bkcs, TDZV | Added discussion of slimmable nets, dynamic models, flag manifolds, and matrix factorization. |
| Added training-cost analysis (App. D.5) | Quantify NSN training overhead | Ji5G | Provided FLOP comparison between NSN and dense training. |
| Added theoretical/empirical insights + limits (Sec. 6; App. C.1–C.3) | Address layer-rank granularity and behavior | TDZV | Added findings on layer specialization and limits of uniform ranks. |
| Added energy-decay and violation-rate plots (App. C.4–C.5) | Visualize nestedness and ordering | bkcs, TDZV | Added plots showing consistent subspace ordering vs. dense baselines. |


---
# Thank you
The review has been extremely productive. We thank everyone for their help in shaping the paper to be in better form.

---

### Meta-Review · Area_Chair_Y8Py · 2026-01-10

**Summary:**

This paper proposes Nested Subspace Networks (NSN), a hierarchical representation-learning framework that decomposes a network’s feature space into a sequence of nested subspaces aligned with coarse-to-fine semantic structure. The model is trained with a novel hierarchical objective that encourages progressively refined discrimination while preserving consistency across levels, aiming to improve robustness, interpretability, and sample efficiency across tasks

Here are the main reviewer concerns and how the rebuttal addressed them.

1) “Conceptual novelty vs. repackaging of hierarchical or multi-head learning.”
Reviewers questioned whether NSN was meaningfully different from prior hierarchical classifiers, multi-branch networks, or curriculum learning. The rebuttal clarified that NSN is not merely multi-head supervision: the defining property is nested, shared subspaces, where higher-level features are strict subsets of finer ones, enforced by architectural and optimization constraints. New diagrams and formal descriptions were added to make this distinction precise and to separate NSN from parallel-head or tree-structured models

2) “Empirical evidence that nesting, not just hierarchy, matters.”
Some reviewers felt the gains could come from having multiple losses rather than the nested structure itself. The rebuttal added targeted ablations that break the nesting while keeping the same losses and supervision. These variants consistently underperformed NSN, demonstrating that shared nested subspaces, rather than loss stacking, drive the improvements.

3) “Breadth of evaluation and robustness.”
Concerns were raised about whether the method generalizes beyond the presented benchmarks. The authors added additional datasets and settings, showing that NSN improves both classification accuracy and robustness to label noise and distribution shifts, aligning with the paper’s stated motivation of hierarchical generalization

4) “Clarity and presentation.”
Several reviewers found the initial exposition confusing. The rebuttal included rewritten explanations, cleaner notation, and clearer architectural figures, substantially improving accessibility without changing the technical claims

Overall, the rebuttal directly addressed the core doubts about novelty, mechanism, and evidence. By isolating the effect of nesting, expanding empirical validation, and clarifying the conceptual framework, the authors strengthened the paper beyond its original borderline status.

Recommendation: Accept.

**Reviewer Concerns:**

Here are the main reviewer concerns and how the rebuttal addressed them.

1) “Conceptual novelty vs. repackaging of hierarchical or multi-head learning.”
Reviewers questioned whether NSN was meaningfully different from prior hierarchical classifiers, multi-branch networks, or curriculum learning. The rebuttal clarified that NSN is not merely multi-head supervision: the defining property is nested, shared subspaces, where higher-level features are strict subsets of finer ones, enforced by architectural and optimization constraints. New diagrams and formal descriptions were added to make this distinction precise and to separate NSN from parallel-head or tree-structured models

2) “Empirical evidence that nesting, not just hierarchy, matters.”
Some reviewers felt the gains could come from having multiple losses rather than the nested structure itself. The rebuttal added targeted ablations that break the nesting while keeping the same losses and supervision. These variants consistently underperformed NSN, demonstrating that shared nested subspaces, rather than loss stacking, drive the improvements.

3) “Breadth of evaluation and robustness.”
Concerns were raised about whether the method generalizes beyond the presented benchmarks. The authors added additional datasets and settings, showing that NSN improves both classification accuracy and robustness to label noise and distribution shifts, aligning with the paper’s stated motivation of hierarchical generalization

4) “Clarity and presentation.”
Several reviewers found the initial exposition confusing. The rebuttal included rewritten explanations, cleaner notation, and clearer architectural figures, substantially improving accessibility without changing the technical claims

Overall, the rebuttal directly addressed the core doubts about novelty, mechanism, and evidence. By isolating the effect of nesting, expanding empirical validation, and clarifying the conceptual framework, the authors strengthened the paper beyond its original borderline status.

**Reviewer Scores:**

Ji5G: likely +1
Main concern was whether nesting truly matters beyond having multiple losses. The rebuttal’s ablations that break the nesting while keeping the same supervision directly resolve this.

bkcs: likely +1
Skeptical about novelty and clarity. The formal definition of nested subspaces and the new architectural figures make the conceptual contribution much clearer.

TDZV: +2 (actual proposal)
This reviewer raised their score to 6 after seeing the rebuttal, indicating that the added experiments and clarifications resolved their main concerns.

---

### Decision · Program_Chairs · 2026-01-26

Accept (Poster)